

# Interactions of fertilisation and crop productivity on soil nitrogen cycle microbiome and gas emissions

Laura Kuusemets[1*], Ülo Mander[1], Jordi Escuer-Gatius[2], Alar Astover[2], Karin Kauer[2], Kaido Soosaar[1], Mikk Espenberg[1]

[1]University of Tartu, Institute of Ecology and Earth Sciences, Vanemuise 46, Tartu, 51003, Estonia
[2]Estonian University of Life Sciences, Institute of Agricultural and Environmental Sciences, Kreutzwaldi 5, Tartu, 51014, Estonia

*Correspondence to*: Laura Kuusemets (laura.kuusemets@ut.ee)

**Abstract.** Fertilised soils are a significant source of nitrous oxide ($N_2O$), a highly active greenhouse gas and stratospheric ozone depleter. Nitrogen (N) fertilisers, while boosting crop yield, also lead to $N_2O$ into the atmosphere, impacting global warming. We investigated relationships between mineral N fertilisation rates and additional manure amendment with different crop types through the analysis of abundances of N cycle functional genes, soil $N_2O$ and $N_2$ emissions, nitrogen use

efficiency (NUE), soil physicochemical analysis and biomass production. Our study indicates that $N_2O$ emissions are predominantly dependent on the mineral N fertilisation rate and enhance with increased mineral N fertilisation rate. Higher $N_2O$ emissions were attained with the application of manure. Manure amendment also increased the number of N cycle genes that are significant in the change of $N_2O$. Contrary to our hypothesis, there was no significant influence of crop type on soil $N_2O$ emissions. The study indicated dominance of nitrification over denitrification in the soil. Microbial analyses also

showed the potential role of comammox and DNRA processes as a source of $N_2O$. Our study did not find soil moisture to be significantly linked to $N_2O$ emissions. Results of the study provide evidence that for wheat, a fertilisation rate of 80 kg N ha$^{-1}$ is closest to the optimal rate for balancing biomass yield, $N_2O$ emissions, and achieving high NUE. Sorghum showed potential for cultivation in temperate climate, as sorghum maintained low $N_2O$ emissions and N losses on mineral N fertilisation rate of 80 kg N ha$^{-1}$.

## 1 Introduction

The rising demand for agricultural commodities and the management of agroecosystems are important factors contributing to global environmental problems. Increasing crop yield while reducing pollution from agricultural production is crucial

(Abdalla *et al*., 2019; Tilman *et al*., 2011). Global food demand projections suggest a 50% increase in agricultural production by 2050 (compared to 2012) to feed the fast-growing human population (FAO, 2017). Enhancing agricultural production involves actions such as expanding agricultural land, applying more fertilisers, and using water resources and



fertilisers more effectively (Tian *et al*., 2021). The yield of main crops like maise, rice, wheat, and soy are projected to decline globally in present agricultural regions due to climate system changes (Asseng *et al*., 2015; Liu *et al*., 2016; Ostberg

*et al*., 2018). By the 2080s, developing countries, often experiencing temperatures near or above crop tolerance levels, are expected to see a 10-25% decrease in agricultural productivity due to global warming. In contrast, developed countries, many with lower mean temperatures, are predicted to experience a 6% decrease to an 8% increase in productivity (Mahato *et al*., 2014). Fertilised soils are a significant source of nitrous oxide ($N_2O$), contributing to the greenhouse effect and ozone depletion (Ravishankara *et al*., 2009; Shcherbak *et al*., 2014). $N_2O$ has 273 times higher global warming potential than

carbon dioxide ($CO_2$) over a 100-year timescale (IPCC, 2021).

Nitrogen (N) fertilisers enhance plant productivity as plants use N to produce proteins for cell construction, a building block for DNA, and a significant component of chlorophyll (photosynthesis) (Andrews and Lea, 2013; Kaur *et al*., 2017). However, the applied N with fertilisation is often excessive for plant needs (Robertson and Vitousek *et al*., 2009; Zhou *et al*., 2016). About half of the applied N to the fields is not taken up by crops (Coskun *et al*., 2017); which may lead to N loss in

the surrounding environment. This results in adverse ecological impacts, such as eutrophication of aquatic ecosystems and increased gaseous emissions of N into the atmosphere (Cameron *et al*., 2013; Liu *et al*., 2017; Whetton *et al*., 2022). Even without adding N fertiliser in the current season or year, background $N_2O$ emissions (BNEs) may still occur. BNEs are caused by different N sources, including residual N in the soil from previous years' N application, deposition from the atmosphere, biological $N_2$ fixation and mineralised N from plant residues (Gu *et al*., 2007; Kim *et al*., 2013, Abdalla *et al*.,

50 2022).

The key microbial processes leading to soil N loss are nitrification and denitrification (Thomson *et al*., 2012). In agriculture, 70% of N fertilisers added to the soil are lost due to these processes (Saud *et al*., 2022). Nitrification was traditionally viewed as a two-step process carried out by separate functional groups of microorganisms, oxidising ammonium ($NH_4^+$) sequentially to nitrite ($NO_2^-$) and nitrate ($NO_3^-$) under aerobic conditions (Kuypers *et al*., 2018; Koch *et al*., 2019; Nardi *et*

*al*., 2020). However, in 2015, a significant advancement in our understanding of nitrification occurred with the discovery that a single microorganism, through the comammox (complete ammonia oxidation) process, can perform both nitrification steps (Daims *et al*., 2015; Van Kessel *et al*., 2015).

Nitrification can reduce N availability for plant uptake by up to 50%, primarily due to $NO_3^-$ leaching and $N_2O$ emissions (Beeckman *et al*., 2018). Denitrification is a microbially-catalysed process under oxygen-limited condition responsible for

transforming $NO_3^-$ sequentially to gaseous forms of N: nitric oxide, $N_2O$ and atmospheric N (Philippot *et al*., 2007; Zaman *et al*., 2012). The input of N fertilisers affects the soil's mineral N pool by providing larger amounts of available N for nitrification and denitrification processes, contributing to $N_2O$ emissions (Engel *et al*., 2010).

Dissimilatory nitrate reduction to ammonium (DNRA) contributes $NH_4^+$ to the soil for biological production as a N fertiliser, conserving bioavailable N in the soil and preventing the leaching of $NO_3^-$ (Bai *et al*., 2020; Pandey *et al*., 2020). DNRA

competes with denitrification in $NO_3^-$-reducing processes, both requiring $NO_3^-$ (Putz *et al*., 2018). Similarly to denitrification and nitrification processes, DNRA can also be a source of $N_2O$, although the quantities are modest (Rütting *et*



*al.*, 2011; Stremińska *et al.*, 2012; Zaman *et al.*, 2012). These processes are mediated by different functional marker genes, including archaeal, bacterial and comammox *amoA* genes for nitrification, *nrfA* genes for DNRA and *nosZ* clad I and II, *nirK, nirS* genes for denitrification (Zaman *et al.*, 2012; Hu *et al., 2015;* Zhang *et al.,* 2021).

C3 photosynthesis, a dominant pathway among plants and found in wheat and barley, uses the Calvin-Benson pathway, while an alternative the Hatch-Slack pathway is used by C4 plants like sorghum and maise (Hibberd and Quick, 2002; Ehleringer and Cerling, 2002; Ehleringer, 1979; Ledvinka, 2022). In C3 plants, water loss through transpiration during $CO_2$ uptake is a risk in hot and water-limited conditions (Joshi *et al*, 2022; Stevens *et al.*, 2022). However, C4 plants, with higher water use efficiency and greater tolerance to hot and dry environments, make the cultivation of sorghum and other drought-

tolerant plants likely to expand in regions affected by droughts (Anderson *et al.*, 2020). Due to climate change, sorghum, as a resilient plant, is considered a novel crop for temperate Europe (Schaffasz *et al.*, 2019).

Previous studies on long-term fertilisation experiments have mostly focused on fertilisation's yield effects and changes in soil organic matter (Cvetkov and Tajnšek, *et al.*, 2009; Hijbeek *et al.*, 2017; Káš *et al.*, 2010; Spiegel *et al.*, 2010; Tajnšek *et al.*, 2013). Improved management of arable soils holds significant potential for mitigating greenhouse gas emissions, as

agroecosystems contribute ca 66% of total anthropogenic $N_2O$ emissions (Davidson and Kanter, 2014; Paustian *et al.*, 2016; Shen *et al.*, 2021). Efficient mitigation of N loss requires a comprehensive understanding of microbial processes related to $N_2O$ emissions in agricultural soils (Davidson and Kanter, 2014; Shen *et al.*, 2021).

The general objectives of the study were to evaluate temporal patterns of gaseous N loss, link mechanistic process understanding based on abundances of functional N cycle genes in arable mineral soil, and evaluate the performance of

different crops (including novel crop in Northern Europe) in terms of biomass production and $N_2O$ emissions under mineral and organic fertilisation. The following hypotheses were tested: (1) crop type significantly affects $N_2O$ emissions; (2) nitrification is the primary pathway of soil $N_2O$ production due to aerobic conditions; (3) in arable mineral soil, low soil moisture decreases $N_2O$ losses; (4) amendment of manure fertiliser increases soil $N_2O$ emissions and affects the soil microbial community; (5) sorghum (*Sorghum bicolor x Sorghum sudanense*) is a prospective crop to cultivate in temperate

climate.

## 2 Material and methods

### 2.1 Field experiment description

The field study was conducted on the IOSDV (Internationaler Organischer Stickstoff Dauerdüngungs Versuch, International

Organic Nitrogen Long-term Fertilisation Experiment) Tartu experimental field. The experiment was set up as a three-field crop rotation experiment in 1989 to investigate the long-term effects of mineral and organic fertilisation on crop types and soil properties. Initially, the crop rotation was potato–spring wheat–spring barley (Astover *et al.*, 2016). In 2019, potato was replaced with sorghum-sudangras hybrid. The experimental site is located in Tartu, southern Estonia, Northern Europe



(58°22'30" N, 26°39'48" E). In 2022, the average temperature in the area was −2.0 °C in winter, 4.6 °C in spring, 18.1 °C
in summer and 7.2 °C in autumn. The mean annual precipitation was 531 mm (Republic of Estonia Environment Agency,
2023) in 2022. The soil type is *Stagnic Luvisol* combined with *Fragic Glossic Retisol* (IUSS WG WRB 2015). The soil has a
sandy loam texture and the thickness of the humus layer is 27-32 cm.

The experiment was organised into 12 plots in a systematic block design (Figure 1) with three replications. Every plot was
50 m$^2$ in size. The crop species studied were spring barley (cultivar "Elmeri"), sorghum (*Sorghum bicolor x Sorghum
sudanense,* cultivar "SUSU"), and spring wheat (cultivar "Mistral"). The fertiliser treatment consisted of mineral N
fertilisation and mineral fertilisation with farmyard manure amendment. Three mineral N fertiliser treatment rates were
studied: 0, 80 and 160 kg N ha$^{-1}$). The mineral fertiliser applied was ammonium nitrate ($NH_4NO_3$). The farmyard manure
rate added to the plots with manure amendment was 40 t ha$^{-1}$ of manure (231.2 kg N ha$^{-1}$). The main management activities
and timing in the field are displayed in Table S1 in Supplementary materials.

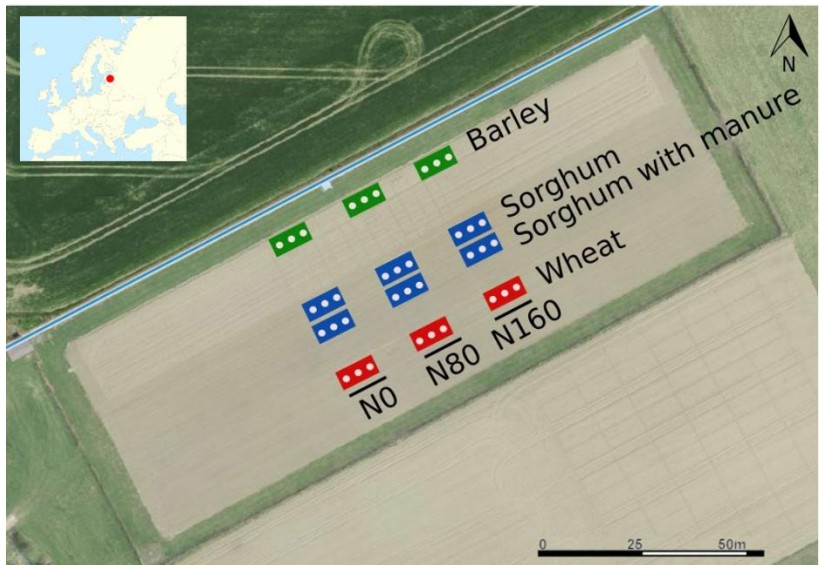

**Figure 1:** Satellite view of the study site with study plots (from Maa-amet). Each plot constituted of three sampling spots
indicated as white dots. N0 – 0 kg N ha$^{-1}$, N80 – 80 kg N ha$^{-1}$, N160 – 160 kg N ha$^{-1}$ as mineral fertiliser.

### 2.2 Gas sampling for N₂O flux analyses

The field study was conducted during the growing season from April 2022 to October 2022. Sampling took place on 15
different dates, starting on April 27[th] and ending on October 12[th] (every week until the end of June and then twice a month
until the end of September). Gas samples for N₂O flux analysis were collected on all fifteen fieldwork days. N₂O gas
sampling was carried out using the static chamber method (Hutchinson and Livingston, 1993). Polyvinyl chloride chambers
(Ø 50 cm, volume 65 l) were placed on top of the collars during the gas sampling. Chamber extensions were used for some



treatments of sorghum on four occasions as the chambers alone were too low/small to accommodate the growing crops. Pre-vacuumed 50 ml glass vials were used for gas sampling. Gas samples were collected at 20 minutes intervals for one hour (0, 20, 40, 60 min). The concentration of $N_2O$ in the collected air was measured in the Biogeochemical Cycling Research Laboratory in the Department of Geography, University of Tartu, with the gas chromatograph Shimadzu GC-2014 (Kyoto, Japan), equipped with electron capture and flame ionisation detectors.

## 2.3. Soil sampling and physicochemical analyses

Soils were sampled for chemical and microbiological analyses six times (April 27th, May 9th, June 2nd, July 7th, September 2nd, October 12th). Soil sampling was conducted after gas sampling. Soil samples were collected close to collars with a soil probe from the top 10 cm of the soil. Until chemical and microbiological analyses, samples were stored at +4 ˚C and −20 ˚C, respectively. In addition to soil sampling, temperature (°C) at a 10 cm depth, moisture ($m^3/m^3$), and electrical conductivity (dS/m) of soil were measured. The soil samples were analysed for total carbon ($C_{tot}$), total nitrogen ($N_{tot}$), nitrate-nitrogen ($NO_3^--N$), and ammonium-nitrogen ($NH_4^+-N$) concentrations in the Soil Science and Agrochemistry Laboratory of Estonian University of Life Sciences. $N_{tot}$ and $C_{tot}$ analyses were done by Dumas method with dry combustion on a VarioMAX CNS elemental analyser (ELEMENTAR, Elementar Analysensysteme GmbH, Langenselbold, Germany). $NO_3^--N$ analyses were done according to EPA (United States Environmental Protection Agency) method 9056: determination of inorganic anions by ion chromatography. $NH_4^+-N$ analyses were done according to Thermo Fisher Application Note 141 (AU204: Determination of Inorganic Cations and Ammonium in Environmental Waters Using a Compact Ion Chromatography System) using ion chromatography.

The hot-water extractable C (HWEOC) represents the readily mineralising carbon (C) fraction and was determined on dry soil samples by a modified method of Haynes and Francis (1993) in two steps.

In the first step the soil was shaken with deionized water at room temperature for 1 h. After that the soil suspension put into the thermostat at 80 °C for 16 h. The mixture was centrifuged for 10 min at 8000 rpm and filtered through a 0.45-μm membrane filter (25-mm diameter, nylon, Agilent®). The HWEOC concentration was determined from the extracts by the VarioMaX CNS analyzer (ELEMENTAR, Elementar Analysensysteme GmbH, Langenselbold, Germany).

## 2.4 Total biomass

The total biomass (above- and below-ground) was measured at the maturity phase. The above-ground biomass was cut from the ground level in a 0.2 $m^2$ area near each collar. The belowground biomass samples were taken with a soil auger (Ø 34 cm). Frasier et al. (2018) provides a more detailed description of the method used for below-ground biomass measurement.The sampling depth extended to the plowing depth, where most of the roots are found, up to a depth of 18 cm. Samples were stored at +4 °C until the roots were washed on a sieve (mesh size 0.5 mm).





Dry matter yield was determined after drying the biomass (including roots) at 70 °C to constant weight. The straw and grains were separated before weighing as air dry.

The biomass (straw, grain, roots) were milled and the $N_{tot}$ content was determined by the Dumas method with dry combustion on a VarioMAX CNS elemental analyser (ELEMENTAR, Elementar Analysensysteme GmbH, Langenselbold, Germany).

## 2.5 Soil microbial analyses

### 160 2.5.1 DNA extraction

DNA was extracted from 0.25 g of soil samples using the DNeasy® PowerSoil® Pro Kit (Qiagen, Hilden, Germany) following the manufacturer's instructions. The difference from the instruction was the homogenisation of samples with homogeniser, Precellys 24 (Bertin Technologies, Montaigne-le-Bretonneux, France), for 20 s at the rate of 5000 rpm. The concentration and quality of the extracted DNA were evaluated with an Infinite 200 M spectrophotometer (Tecan AG,
Männedorf, Switzerland). The extracted DNA was stored in a freezer at −20 °C.

### 2.5.2 Quantification of gene copies using qPCR

Quantification of the 16S rRNA genes of bacteria and archaea, along with the quantification of nitrification (bacterial, archaeal, and comammox *amoA*), denitrification (*nirS*, *nirK*, *nosZI*, and *nosZII*) and DNRA (*nrfA*) genes was done using
quantitative polymerase chain reaction (qPCR). qPCR reactions were performed by The Rotor-Gene Q thermocycler (Qiagen). The reaction mixture of 10 ml consisted of extracted DNA (1 ml), gene-specific forward and reverse primers, Maxima SYBR Green Master mix reagent (5 ml; Thermo Fisher Scientific, Waltham, MA, USA) and distilled water. Each sample was amplified two times. All of the qPCR assays included two DNA-free negative control samples. Details on thermal cycling conditions and used primers are added in Table S2 in Supplementary Materials. The Rotor-Gene® Q
software v. 2.0.2 (Qiagen) and LinRegPCR v. 2020.2. were used to assess the qPCR results. The amount of gene copies was calculated using standard curve ranges, and results were presented in gene copies per gram of dry matter (copies/g dw). Espenberg *et al*. (2018) provides a more detailed description of the used qPCR methodology.

## 2.6 Statistical analyses and modelling

Statistical software programs Statistica (v. 7.1) and R (v. 4.0.4) were used for statistical analysis and visualising the data. Principal component analysis (PCA) were conducted on soil physicochemical parameters and microbiological data (abundance of functional marker genes) with the "FactoMineR" (Lê *et al.,* 2008) and "factoextra" (Kassambara *et al*., 2020)



packages in the software R. Analysis of variance (ANOVA) with post-hoc Tukey HSD test was used to find statistically significant differences between different fertilisation rates, use of manure and crop types. Spearman's rank correlation coefficient measured the association between $N_2O$ emissions and gene abundances and environmental factors. Random forest analysis was conducted using Boruta v. 8.0 (Kursa and Rudnicki, 2010) to identify the gene parameters that best predicted $N_2O$ fluxes.

Nitrogen use efficiency (NUE, kg DM $kg^{-1}$ $N^{-1}$) was calculated as the biomass yield produced per unit of N applied (Pandey *et al*., 2001; Supplementary Methodology S2). The $N_2$ emissions were estimated from the measured $N_2O$ emissions using the $N_2:N_2O$ ratio, which was calculated as proposed in the DAYCENT model (Parton *et al*., 2001), with the equations described in Del Grosso *et al*. (2000) (Supplementary Methodology S1), where the ratio $N_2:N_2O$ is a function of the content of $NO_3^-$ in the soil, $CO_2$ emissions, and water-filled pore space (WFPS). The change of soil N content (kg N $ha^{-1}$) was calculated according to Sainju, 2017 as the difference between the initial and final soil total N contents (Supplementary Methodology S3). N losses are calculated by substracting N outputs and change of soil N content from N inputs (Sainju *et al*., 2017; Escuer-Gatius *et al.*, 2022; Supplementary Methodology S4)

## 3 Results

### 3.1 Soil physicochemical characteristics and biomass production

The $NH_4^+$-N content in soil decreased on most of the plots at the beginning of the study period, while $NO_3^-$-N content in the soil was increasing (Supplementary Figure S5). Fertilised plots had higher soil $N_{tot}$, $C_{tot}$, $NO_3^-$-N and $NH_4^+$-N content compared to non-fertilised plots according to the principal component analysis (PCA) (Figure 2). For sorghum without manure amendment plots (Figure 2C), $NO_3^-$-N and $NH_4^+$-N contents were more different from each other compared to sorghum with manure amendment plots, where $NO_3^-$-N and $NH_4^+$-N contents were relatively similar (Figure 2D). HWEOC concentrations were higher in sorghum plots with farmyard manure amendment compared to sorghum plots without manure amendment.

Soil moisture ranged from 0.02 $m^3/m^3$ to 0.32 $m^3/m^3$ with an average of 0.23 $m^3/m^3$ over the study period (Supplementary Figure S6). There were no significant correlations between soil moisture and $N_2O$ emissions. Over all crop types, soil moisture was not significantly linked to gene copy numbers, except *nirS* (Supplementary Table S6).





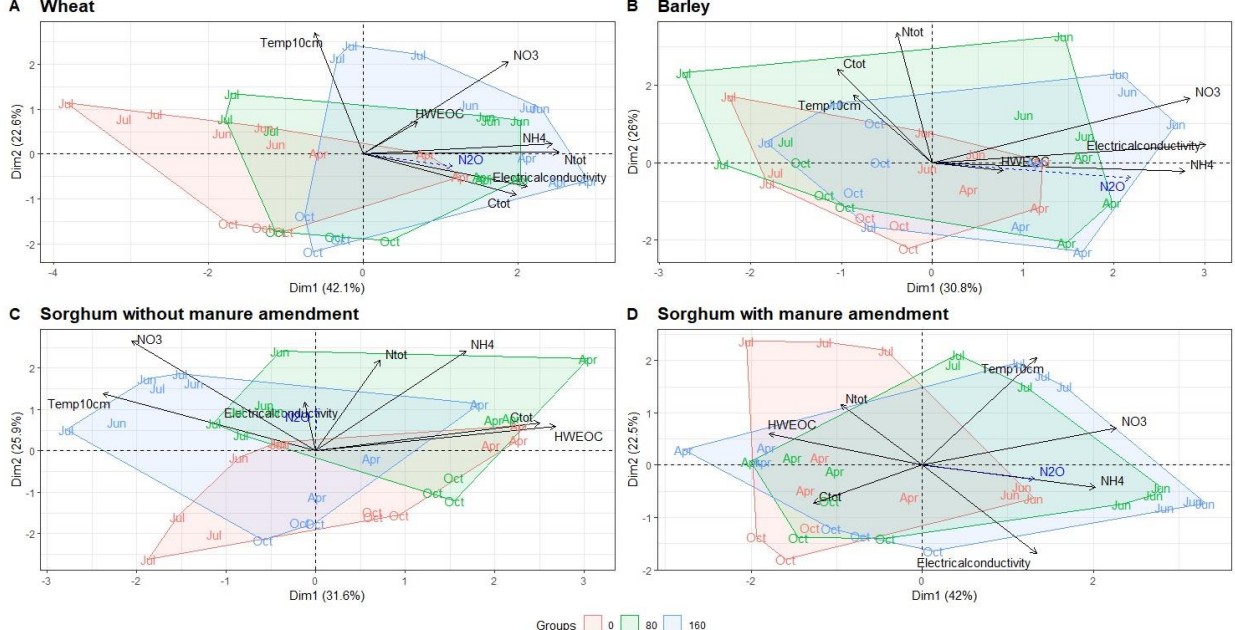


**Figure 2:** Principal components analysis (PCA) ordination plots demonstrate the grouping of fertilisation rates according to physicochemical parameters for different crop type. N$_2$O is added as a supplementary variable. The month indicates the sampling time. Abbreviations: Ctot – total carbon content of soil; Ntot – total nitrogen content of soil; HWEOC – hot-water extractable organic carbon.


The total dry biomass of barley ranged between 2.6 to 6.4 t ha$^{-1}$, and wheat between 4.6 to 8.5 t ha$^{-1}$ depending on the mineral N fertilisation rate (Figure 3). For sorghum without manure amendment, the total dry biomass varied between 2.3 and 7.1 t ha$^{-1}$, and for sorghum with manure amendment, the total dry biomass varied between 8.2 and 11.7 t ha$^{-1}$.

The biomass production was higher per unit area of crop growth with higher fertiliser input (Figure 3A). Total biomass was

significantly positively correlated with N$_{tot}$ ($p<0.01$), C$_{tot}$ ($p<0.05$) and NO$_3^-$-N ($p<0.001$) levels in soil (Supplementary Table S4). Also, higher N fertilisation rate increased N concentrations in the crop biomass (Figure 3B).





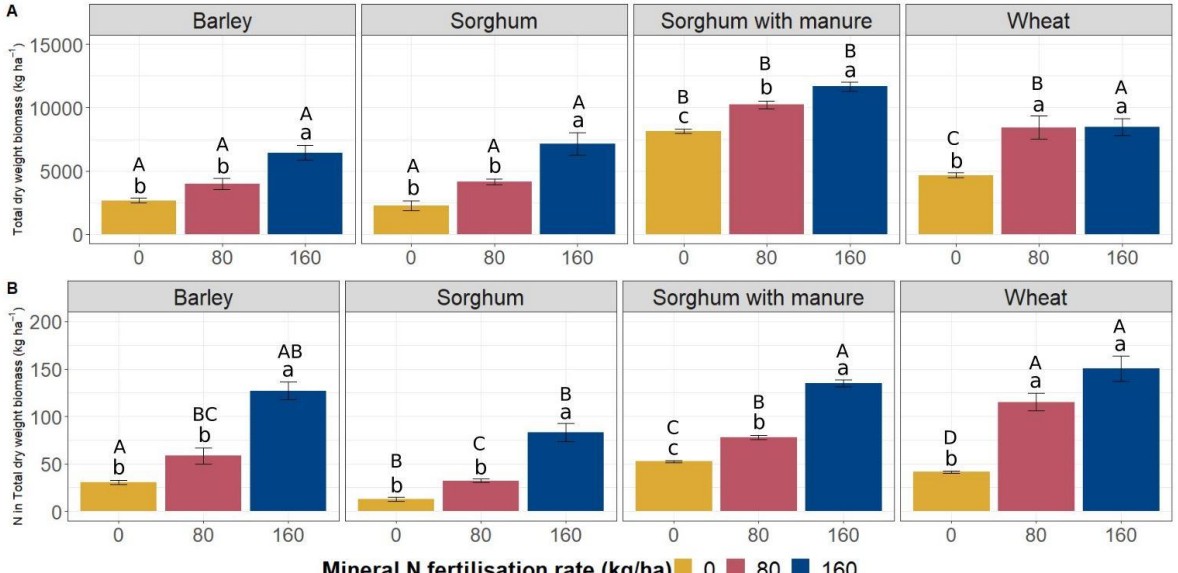

**Figure 3:** Total dry weight biomass (aboveground + belowground) produced per unit area according to crop types and
fertilisation rates. Error bars show standard errors. Letters above the boxes indicate statistically significant differences at the
significance level p < 0.05. Lowercase letter indicate comparisons inside crop type, while uppercase letters indicate
comparisons of the same fertilisation rate over all crop types.

The highest values of nitrogen use efficiency (NUE) were obtained from wheat plots, and the lowest from sorghum plots.
The average NUE of wheat plots at fertilisation rate 80 was 0.84, and at rate 160, it was 0.64. For sorghum plots with manure
amendment, NUE at mineral N fertilisation rate 0 was 0.15, at rate 80 was 0.16, and at rate 160 was 0.25. For sorghum plots
without manure amendment, the average NUE at fertilisation rate 80 was 0.12, and at rate 160, it was 0.25. The NUE for
barley plots at fertilisation rate 80 was 0.35, and at fertilisation rate 160, it was 0.45. The highest estimated N losses occurred
on sorghum plots with manure amendment (Supplementary Table S3). In general, wheat plots at different fertilisation rates
lost more N compared to sorghum plots without manure amendment. The lowest estimated N losses occurred on barley plots.

**3.2 Nitrogen cycle genes**

The abundances of N cycling genes on plots with different fertilisation rates and crop species show different patterns
throughout the study period (Supplementary Figures S1, S2, S3, and S4). The principal component analysis (PCA) of the N
cycle genes abundances showed differences between sites with different fertilisation rates (Figure 4). There were greater
differences in gene abundances between three different mineral N fertilisation rates (Figure 4) in all sorghum plots compared
to barley and wheat plots. For sorghum without manure amendment (Figure 4C), archaeal 16S rRNA and *nosZII* gene
abundances were highest for fertilisation rate 80, but for sorghum with manure amendment (Figure 4D), the highest archaeal




16S rRNA and *nosZII* gene abundances were for fertilisation rate 160. For all sorghum plots, comammox *amoA* gene

abundance was highest on non-fertilised plots. However, fertilised wheat and barley plots had higher comammox *amoA* gene abundance compared to non-fertilised plots.

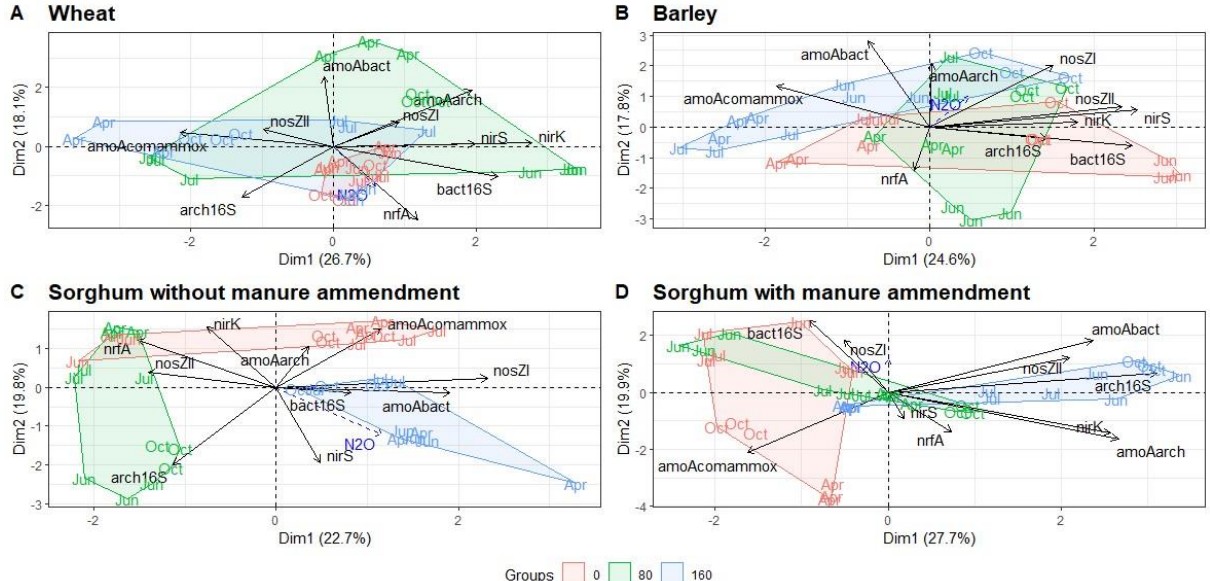

**Figure 4:** Principal components analysis (PCA) ordination plots demonstrate the grouping of fertilisation rates according to

functional marker genes abundances for different crop type. $N_2O$ is added as a supplementary variable. The month shows the sampling time. Abbreviations: bact16S – bacterial 16S rRNA gene; arch16S – archaeal 16S rRNA gene; amoAbact – bacterial *amoA* gene; amoAarch – archaeal *amoA* gene; amoAcomammox – comammox *amoA* gene.

### 3.3 N2O emissions

The $N_2O$ emissions over the course of the study period show that different fertilisation rates influence $N_2O$ emissions, and the highest $N_2O$ emissions tend to be emitted from the highest N fertiliser treatment (160 kg N ha$^{-1}$) (Figure 5). $N_2O$ emissions among all crop species tended to be higher during the first part of the study period (spring and early summer). Taken together, the highest average $N_2O$ emissions for barley plots were measured in the middle of May, for sorghum plots without and with manure in the middle of June, and for wheat plots at the beginning of June.




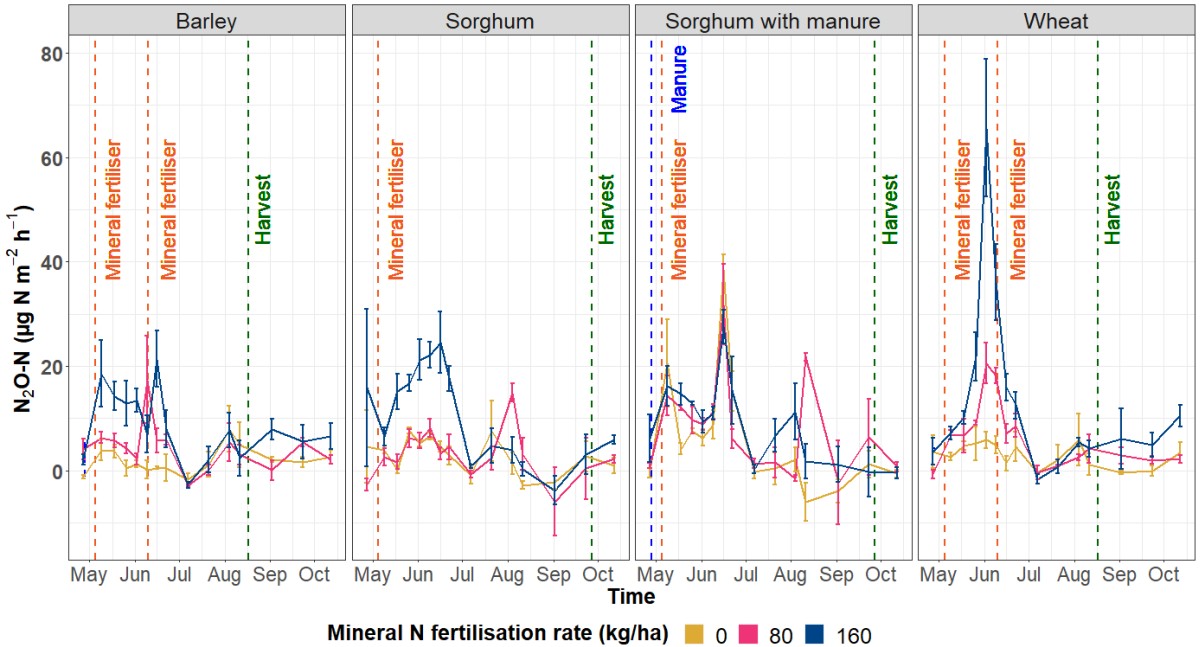

**Figure 5:** $N_2O$ emissions ($\mu$g N m$^{-2}$ h$^{-1}$) according to crop types and fertilisation rates during the study period.

Cumulative $N_2O$ and $N_2$ emissions over the study period show that plots with barley, wheat and sorghum without manure

have the highest emissions emitted from the highest fertilisation rate (Figure 6A, B). For wheat and barley plots, there is a clear pattern of increasing $N_2O$ emissions with increasing fertilisation rates.

For barley plots, cumulative $N_2O$ emissions did not significantly differ between fertilisation rates 0 and 80 (Figure 6A). However, $N_2O$ emissions on barley plots were significantly higher at fertilisation rate 160 than at rate 0 and 80 ($p < 0.05$). Similarly, for wheat plots, cumulative $N_2O$ emissions were also significantly higher at fertilisation rate 160 compared to rates

0 ($p < 0.05$) and 80 ($p < 0.05$); however, fertilisation rates 0 and 80 did not significantly differ from each other. For plots with sorghum without manure, cumulative $N_2O$ emissions at fertilisation rate 160 were significantly higher compared to fertilisation rates 0 ($p < 0.05$) and 80 ($p < 0.05$). For sorghum with manure plots, cumulative $N_2O$ emissions from the three different fertilisation rates did not differ significantly from each other.

For barley plots, the cumulative $N_2$ emissions were significantly higher at fertilisation rate 160 compared to rates 0 ($p < 0.05$)

and 80 ($p < 0.05$) (Figure 6B). For wheat, sorghum with and without manure plots, cumulative $N_2$ emissions emitted from all three fertilisation rates did not significantly differ from each other.



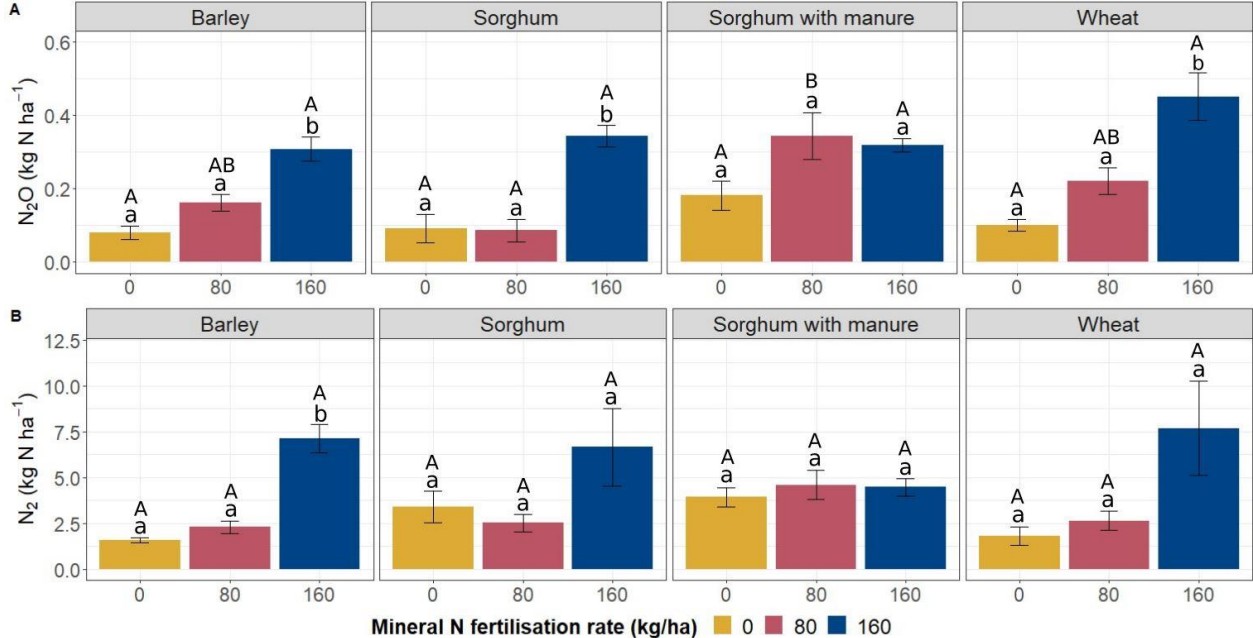

**Figure 6:** Cumulative $N_2O$ and $N_2$ emissions according to crop types and fertilisation rates. Error bars show standard errors.
Letters above the boxes indicate statistically significant differences at significance level $p < 0.05$ according to a post-hoc Tukey HSD test. Lowercase letters indicate comparisons within crop types. Uppercase letters indicate comparisons of the same fertilisation rate over all crop types.

## 3.4 Relationships between environmental and genetic parameters and N2O emissions

Mineral N fertilisation rate ($p<0.001$) and manure amendment ($p<0.01$) significantly influenced $N_2O$ emissions (Table 1). Crop type did not significantly influence $N_2O$ emissions. The effect of mineral N fertilisation rate on $N_2O$ emissions ($\omega^2 = 0.528$) was larger compared to effects of crop type ($\omega^2 = 0.021$) and manure amendment ($\omega^2 = 0.121$).



**Table 1:** Results of ANOVA and effect size ($\omega^2$) testing the effects of crop type, mineral N fertilisation rate and manure amendment on cumulative $N_2O$ fluxes. Significance is indicated as *** – 0.001; ** – 0.01; * – 0.05; ns – not significant.

| | Df | Sum of Squares | Mean Square | F value | Pr (>F) | $\omega^2$ |
|---|---|---|---|---|---|---|
| Crop type | 2 | 0.350 | 0.175 | 0.957 | 0.39544 | 0.021 |
| Mineral N fertilisation rate | 2 | 8.772 | 4.386 | 23.995 | $5.97 \times 10^{-7}$ *** | 0.528 |
| Manure amendment | 1 | 2.014 | 2.014 | 11.020 | 0.00237 ** | 0.121 |
| Residuals | 30 | 5.483 | 0.183 | | | |

Feature selection algorithm for the $N_2O$ emissions from wheat plots considered bacterial *amoA*, archaeal *amoA*, *nosZI* and

300   *nosZII* genes relevant (Figure 7). For barley plots, bacterial *amoA*, comammox *amoA*, bacterial 16S rRNA, *nirK*, *nirS* and *nosZII* were deemed as important genes in the change of $N_2O$ emissions. For sorghum without manure amendment plots, bacterial *amoA*, comammox *amoA*, archaeal 16S rRNA and *nirK* genes were considered important for the $N_2O$ emissions. For sorghum with manure amendment plots, archaeal *amoA*, bacterial *amoA*, comammox *amoA*, *nirK*, *nirS*, *nosZII*, *nosZI* and *nrfA* genees were considered important for the $N_2O$ emissions.

305

2000



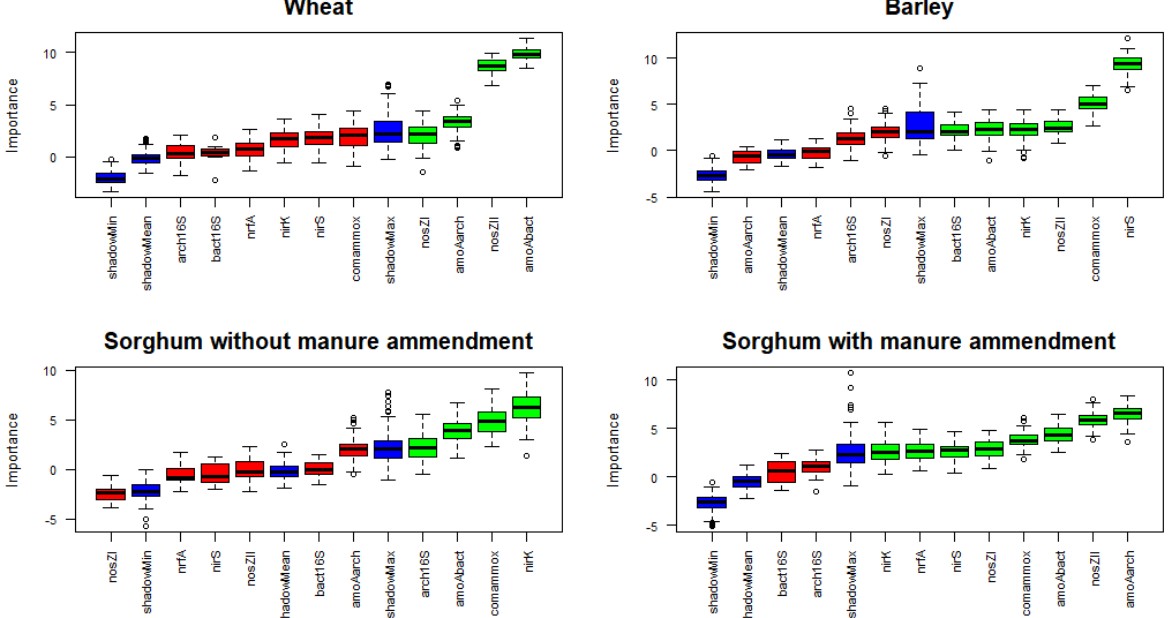

**Figure 7:** Results of feature selection in predicting the genes that are important in the change of N₂O emissions. Important factors are indicated in green, unimportant factors in red, and shadow variables (the random shadow copies of features (noise) will be created to test the feature against those copies to determine if it is better than the noise, and therefore significant) in blue. Abbreviations: bact16S – bacterial 16S rRNA gene; arch16S – archaeal 16S rRNA gene; amoAbact – bacterial *amoA* gene; amoAarch – archaeal *amoA* gene; amoAcomammox – comammox *amoA* gene.

The relationships between gene abundances and N₂O emissions showed that the ratio of *amoA*/*nir* was in a significant positive correlation with N₂O emissions (ρ=0.20; p<0.001). Furthermore, the ratio of *nosZ*/*nir* was also significantly positively correlated with N₂O emissions (ρ=0.21; p<0.001). Archaeal 16S rRNA genes were positively correlated with N₂O emissions over all crop species plots (ρ=0.18; p<0.05). Bacterial *amoA* genes were negatively correlated to N₂O emissions on wheat plots (ρ=−0.40; p<0.05). In addition, comammox *amoA* genes were negatively correlated with N₂O emissions on plots with sorghum without manure (ρ=−0.47; p<0.01). *nirS* genes were positively correlated with N₂O emissions over all crops species plots (ρ=0.19; p<0.05). N₂O emissions from barley plots had a strong positive correlation with *nirS* gene abundance (ρ=0.58; p<0.001). *nosZII* genes were positively correlated with N₂O emissions on plots with sorghum with manure amendment (ρ=0.41; p<0.05). On wheat plots, *nosZII* genes were negatively correlated with N₂O emissions (ρ=−0.46; p<0.01). The correlation matrix is provided as Table S5 in Supplementary materials.



## 4 Discussion

Mineral N fertilisation positively influenced biomass increase in all three crop types (Figure 3A), with similar findings observed in other IOSDV experiments by Csitári *et al*. (2021) and Tajnšek *et al*. (2013). The results also showed a
significant positive correlation between biomass and soil $NO_3^--N$ and $N_{tot}$ content, explaining higher biomass production in fertilised soil, as N limitation is the most influencial factor constraining crop growth (Mengel and Kirkby, 2001). Furthermore, increasing mineral N fertilisation led to higher N accumulation in the biomass (Figure 3). Applying N at higher rates than necessary for optimal yield can increase crop protein content (Serret *et al*., 2008; Mengel and Kirkby, 2001).

The sorghum plots without fertilisation yielded 2.3 t ha$^{-1}$, while those with only manure amendment produced an additional
5.9 t ha$^{-1}$ of total dry biomass (Figure 3A), consistent with the results from Spiegel *et al*. (2010). The positive effect of manure amendment could be attributed to increased availability of macro- and micronutrients. The beneficial impact of manure was evident even at higher mineral N fertilisation rates. However, meta-analysis by Hijbeek *et al*. (2017), covering 20 long-term experiments (including the IOSDV experimental site used in our study) in Europe, reported that organic input does not necessarily guarantee increased crop yields.

In various ecosystems, N cycle genes have been linked to $N_2O$ emissions (Butterbach-Bahl *et al*., 2013; Espenberg *et al*., 2018; Harter *et al*., 2014). The significant positive correlation between the ratio of *amoA/nir* and $N_2O$ emissions (ρ= 0.20, p<0.001) indicates the dominance of nitrification over denitrification in $N_2O$-producing processes. Additionally, an initial decrease in $NH_4^+-N$ content in soil was observed, suggesting $NH_4^+$ consumption (nitrification) and mineral N uptake by plants (Supplementary Figure S5). A simultaneous increase in $NO_3^--N$ accompanied by a decrease in $NH_4^+-N$ was recorded,
likely resulting from the nitrification production process.

*nirS* genes exhibit a positive correlation with $N_2O$ emissions across all crops species plots (ρ=0.19; p<0.05), suggesting that while nitrification is predominant, denitrification is also evident. This finding aligns with results from several other agricultural studies, which also reported a significant positive correlation between *nirS* genes and $N_2O$ emissions (Castellano-Hinojosa *et al*., 2020; Cui *et al*., 2016). Additionally, the ratio of *nosZ/nir* is significantly positively correlated
with $N_2O$ emissions (ρ=0.21, p<0.001). This correlation may be attributed to $N_2O$ being emitted from other sources, such as nitrification.

For all plots, one or more functional marker genes related to nitrification and denitrification were identified as important in the change of $N_2O$ emissions (Figure 7), emphasizing the significance of both processes in $N_2O$ emissions. Comammox was also recognized as an imporant process in $N_2O$ emissions, except in wheat plots, indicating its potential important role.
Additionally, Li *et al*. (2019) demonstrated an order of magnitude higher abundance of comammox *Nitrospira* clade A compared to ammonia-oxidizing archaea and ammonia-oxidizing bacteria in fertilised agricultural soil.

More functional marker genes show significance in the change of $N_2O$ with manure compared to other treatments (Figure 7), indicating that a greater number of N cycle processes are relevant in plots with manure. Additionally, *nosZI, nosZII* and *nirS* genes were identified as important for sorghum with manure amendment, but not for only mineral fertiliser sorghum plots,



which indicates significance of denitrification in these plots. Previous studies also suggest a higher denitrification potential from manure treatment, highlighting the importance of denitrifying microorganisms in manure-fertilised plots (Clark *et al*., 2012; Wan *et al*., 2023). The increased denitrification rate in manure-amended plots may be due to improved soil water retention promoting denitrification and increased availability of labile C content, which is the energy source for denitrifiers. Our results also support higher labile C content in plots with manure amendment (Figure 2). Furthermore, sorghum with

manure plots were the only plots where the *nrfA* gene was identified as an important gene in N$_2$O emissions, suggesting that manure amendment is likely enhancing the rate of DNRA process.

Agricultural soils typically act as a source of N$_2$O (Davidson and Kanter, 2014), as shown in this study. The three mineral N fertilisation rates investigated influenced N$_2$O emissions, with N$_2$O emissions increasing with higher mineral N application rate for all three crop species (Figure 5, 6A). This can be attributed to higher available N levels with increased fertilisation

rates for processes contributing to N$_2$O emissions (Engel *et al*., 2010), as N$_2$O emissions showed a strong positive correlation with both NO$_3^-$-N and NH$_4^+$-N levels in soil. Prior studies have also highlighted a positive relationship between soil N$_2$O emissions and mineral N content (Sosulski *et al*., 2014; Yao *et al*., 2009; Yuan *et al*., 2022). Furthermore, among the investigated factors, the mineral N fertilisation rate was the primary determinant of cumulative N$_2$O emissions (Table 1), indicating that soil N$_2$O emissions are mainly linked to the excess N added with mineral fertiliser in the cropping system

(Supplementary Table S3).

Our study found that crop type did not significantly affect cumulative N$_2$O emissions, while the effects of mineral N fertilisation rate and manure amendment on cumulative N$_2$O emissions were significant (Table 1). It suggests that N$_2$O emissions from soil are more closely related to the excess N in the cropping system than the crop type. However, previous studies have shown a significant effect of crop type on N$_2$O emissions (Bouwman *et al*., 2002; Kaiser and Ruser, 2000).

Manure amendment significantly impacted N$_2$O emissions (Table 1). Additionally, soil N$_2$O emissions were higher under mineral fertiliser plus manure amendment than with mineral fertiliser alone for sorghum. This can be attributed to the overall higher mineral input of N into the cropping system in mineral fertiliser plus manure plots compared to mineral fertiliser-only plots (231.2 kg N ha$^{-1}$ was added extra), enhancing N$_2$O production. In addition to providing nitrifiable N compounds, manure incorporation improves soil conditions for nitrification and denitrification by increasing moisture and adding C to the

soil (Chadwick *et al*., 2000). While the increase in moisture with manure was not detectable from our study, it may be explained by the slow evolution of soil properties over previous years in the 33-year-long fertilisation experiment. Moreover, manure enhances the activity of soil microbes, oxygen consumption, and the development of anaerobic zones in the soil, favouring denitrification (Akiyama and Tsuruta, 2003).

Sorghum plots with manure amendment exhibited high N$_2$O emissions across all fertilisation rates, with emissions increasing

slower than linearly with the fertilisation rate (Figure 6A). Conversely, N$_2$O emissions for sorghum plots without manure amendment increased exponentially with the rising mineral N fertilisation rate. This pattern aligns with findings from other studies that observed N$_2$O emissions responding exponentially to increasing fertilisation rates (Grant *et al*., 2006; Ni *et al*., 2021; Shcherbak *et al*., 2014; Walter *et al*., 2015). The potential explanation is that N input may be surpassing crop needs, as



N$_2$O emissions often grow exponentially when the applied N exceeds the necessary amount for crops (Van Groenigen *et al.*,
2010; Snyder *et al.*, 2009). This could be attributed to the high NO$_3^-$-N availability in the soil for all crop types in our study,
leading to NO$_3^-$-N accumulation when crop needs are exceeded (Legg and Meisinger, 1982). However, the results suggest
that N$_2$O emissions have a positive linear response to fertilisation rates for barley and wheat plots (Figure 6A). Field studies
by Gregorich *et al.* (2005) and Halvorson *et al.* (2008) reported that N$_2$O emissions tend to show linear growth with
increasing N fertilisation rates when N input is less or matches with the amount needed for maximum yield.

Higher N$_2$O emissions were recorded in the initial phase of the study period when fertilisers were applied in spring and early
summer (Figure 5). The addition of both NO$_3^-$ and NH$_4^+$ contributed available N to the cropping system, promoting
microbial activity in N$_2$O-producing processes and subsequently increasing N$_2$O emissions. The increase in N$_2$O emissions
might also be affected by soil ploughing, which enhances the mineralisation of soil organic N and crop residues, releasing
plant-available nutrients (e.g., P, S, N) and thereby increasing substrate availability for microbial processes generating
gaseous N (Lal *et al.*, 2007).

Soil microbial processes leading to N$_2$O production are influenced by soil water content, indirectly affecting oxygen
availability for nitrification and denitrification processes. The recorded lowest soil moisture contents for barley and wheat
plots on 7th of July (Supplementary Figure S6) likely explain the lowest N$_2$O emissions on that date (Figure 5). Previous
studies on N$_2$O emissions and soil moisture dynamics have reported similar trend (Yamulki *et al.*, 1995; Yuan *et al.*, 2022).
At low soil water content, nitric oxide (NO) is found to be the main soil gaseous N emission instead of N$_2$ and N$_2$O
(Davidson, 1991; Medinets *et al.*, 2015). Dry soils may lead to microorganisms experiencing cell dehydration and increased
soil salinity, hindering soil microbial activity and, therefore, the production of gaseous N emissions. Although our study did
not find significant correlations between soil moisture and N$_2$O emissions or most of the functional marker genes.

Considering climate changes and population growth, N$_2$O management should align with crop yield. Biomass increased with
fertilisation rate (Figure 3), except for wheat plots, where plots with fertilisation rates at 80 kg N ha$^{-1}$ and 160 kg N ha$^{-1}$ had
very similar biomass values. Long-term fertilisation experiments (IOSDV) by Káš *et al.* (2010) achieved highest wheat
yields from N fertilisation rate 160 kg N ha$^{-1}$, but our study shows increasing N$_2$O emissions at higher fertilisation rates
(Figure 6A), suggesting potential overfertilisation. The highest nitrogen use efficiency (NUE) was observed at fertilisation
rate 80 kg N ha$^{-1}$ for wheat (NUE = 0.84), indicating a balance between low N$_2$O emissions and high yield. In India,
Chaturvedi et al. (2006) conducted experiments with N fertilisation rates of 0, 25, 50, 75, 100, and 125 kg N ha$^{-1}$, and
identified the highest N input rate as optimal.

On fertilisation rate 160 kg N ha$^{-1}$, N$_2$O emissions significantly increase compared to lower rates, but this rate also results in
significantly higher total dry biomass (Figure 3; Figure 6). The fertilisation rate 80 kg N ha$^{-1}$ appears optimal with low N$_2$O
emissions and N losses (Supplementary Table S3). However, in sorghum plots without manure amendment, NUE values are
low (160 kg N ha$^{-1}$ NUE = 0.25; 80 kg N ha$^{-1}$ NUE = 0.12).



**5 Conclusions**

The results of our study (part of the 33 year old IOSDV experiment) showed that the mineral N fertilisation rate was the dominant factor determining cumulative $N_2O$ emissions. The study observed an increase in $N_2O$ emissions with an elevated mineral N fertilisation rate, attributed to higher $NO_3^-$-N and $NH_4^+$-N levels in fertilised soil. Higher $N_2O$ emissions were measured during spring and early summer when mineral N fertilisers and farmyard manure was applied. These findings supported our hypothesis of higher $N_2O$ emissions on sorghum plots under mineral fertiliser plus manure treatment compared to only mineral fertiliser treatment. Additionally, the number of N cycle genes significant in the change of $N_2O$ also increased with manure amendment. Barley and wheat plots exhibited a positive linear response to fertilisation rates, while $N_2O$ emissions from sorghum plots without farmyard manure amendment responded rapidly to the highest mineral N fertilisation rate. Contrary to our hypothesis, crop type did not have significant effect on $N_2O$ emissions in this study.

Nitrification dominated over denitrification in $N_2O$ production in mineral arable soil, with potential contributions from comammox and DNRA processes. Plots with manure amendment exhibited a greater impact of N cycle microbial processes on $N_2O$ emissions, compared to plots with other crop types. Soil moisture showed no correlation with $N_2O$ emissions and most of the functional marker gene abundances. Nonetheless, the lowest $N_2O$ emissions and functional marker gene abundances were recorded during periods of low soil moisture, suggesting a decrease in $N_2O$ under such conditions.

For wheat, a high NUE value and low $N_2O$ emissions, coupled with relatively high crop yield, suggest that a fertilisation rate 80 kg N ha$^{-1}$ is optimal. Similarly, in sorghum plots with only mineral N fertiliser amendment, a fertilisation rate 80 kg N ha$^{-1}$ resulted in low $N_2O$ emissions and N losses, positioning sorghum as a potential crop for Northern Europe.

**Author contributions.** ME, AA, and ÜM designed the experiment and developed the methodology. LK and JEG carried out the fieldwork. LK analysed the results, performed data visualization, and wrote the original manuscript. JEG and ME participated in data analyses and assisted with paper editing. All authors were involved in revising the paper for submission and contributed to its improvement.

**Competing interests.** The contact author has declared that none of the authors has any competing interests.

**Acknowledgements.** We thank Triin Teesalu for assisting with soil sampling for chemical analyses. Thanks to Tõnu Tõnutare and Kristi Kõva for conducting soil chemical analysis. We would also like to acknowledge Avo Toomsoo for operating the fertilisation experiment and maintenance of the study field.

**Financial support.** The study was supported by the Estonian Research Council (grants number PRG352 and PRG2032), European Research Council (ERC) under the grant agreement No 101096403 (MLTOM23415R), European Commission



through the HORIZON-WIDERA 'Living Labs for Wetland Forest Research' Twinning project No 101079192 and the
European Regional Development Fund (Centres of Excellence EcolChange, TK131, and AgroCropFuture, TK200).

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
