# Peer review of "Interactions of fertilisation and crop productivity on soil nitrogen cycle microbiome and gas emissions"

_EGUsphere, 2024_

## Author Comment (AC3)

**Anonymous Referee #3:** This work presents a relevant topic, that is soil $N_2O$ emissions management, to understand N fertilization and crop type impact, along with a possible involvement of soil microbiome. Therefore, in my opinion, it is relevant to SOIL aims and scope. Introduction is well constructed, and it highlights the relevance of the study in a broader context. Some modifications in text structure are required. Hypotheses and objectives are clearly stated and coherent with the methodology used, even though some of them might require improvements.

**A:** Thank you for your valuable feedback. We have carefully reviewed your comments and addressed all the critical points raised. Detailed responses are provided below.

1. The biggest problem of this paper is a major lack in appropriately describing the experimental design. In fact, it is necessary to specify which crops immediately preceded the ones tested in the current experiment and which fertilization treatments they received. The reason why it is so important is that this experimental design would not be valid if this wasn't a long-term experiment, as there aren't multiple separate and randomized plots. In fact, I think that what you refer to as "replicates" are only multiple sampling points of one unique plot per treatment. If this lack in methodological information will not be addressed appropriately, I fear that it might seriously undermine the reproducibility of this work. In addition, some information about one of the treatments is lacking (manure amendment).

**A:** Thank you for pointing this out. We have revised the manuscript and added additional information about the experimental design according to reviewers' suggestions.

Study is made on long-term three-field crop rotation experiment established in 1989. All fertilizer treatments have applied continuously from first harvest year in 1990. Manure treatment is amended with solid farmyard manure (ca 40 t ha$^{-1}$) in every third year before sorghum/potato. Last year of manure amendment was in year 2022. The farmyard manure is cattle dung with straw bedding, freely fermented before use 6-8 months in heap. In the Material and Methods section, we already had the following information about the preceded crops: "Initially, the crop rotation was potato–spring wheat–spring barley (Astover *et al*., 2016). In 2019, potato was replaced with sorghum-sudangras hybrid."

We will include the use of linear mixed-effects models in the revised version of the manuscript. We use it to test statistical differences between N emissions of different fertilisation rates in plots with different crop types. We use spatial (different fertilisation rate) and temporal (sampling dates) effects as random effects. This model will help account for both fixed and random effects inside the experimental design, which provides better analysis of data.

2. Results have been described quite clearly, although sometimes too much detail is given about findings that don't have a wide importance. In some graphs I think there are some mistakes in results presentation. In the discussion section there are some problems related to the flowing of the text. In fact, often the description of the same topic is divided into multiple, short paragraphs, thus creating some confusion for the reader. Moreover, some of

the speculations are too strong based on the presented results. Overall, the manuscript has a great potential to be improved, but only if the issue with the experimental design is correctly and extensively addressed, as it is the most serious problem of this work.

**A:** Thank you for the constructive feedback. We have revised the manuscript according to reviewer's feedback. In Discussion section, we have removed the repetitive parts and improved flowing of the text. We have also consolidated paragraphs, where discussion of the same topic was previously divided into multiple paragraphs. We have also revised the figures and the Discussion section. In addition, we smoothed the text where needed.

3. L. 12: $N_2O$ I think the term emission is missing here.

**A:** Done!

4. L.s 16-17 You mean higher compared to the application of mineral fertilizers?

**A:** Yes, we mean in comparison to mineral fertilisation. We have clarified it in the manuscript.

5. L. 19 **Microbial analyses** Could you be more specific here?

**A:** Thank you for pointing this out. We have revised the sentence followingly: "Quantification of nitrogen cycle functional genes also showed the potential role of denitrification, comammox and DNRA processes as a source of $N_2O$."

6. L. 23 **sorghum** It is not necessary to repeat the term sorghum here again.

**A:** Done!

7. L. 33 **maise** There is a spelling mistake here.

**A:** Done! We have corrected this error throughout the entire manuscript.

8. L. 34 **in present agricultural regions due to climate system changes** I am sorry, but this is not very clear to me. Could you please rephrase it.

**A:** We have removed this sentence from Introduction section to avoid too verbose text and redundancies.

9. L. 38 This sentence does not fit well in this paragraph. I think it is better to move it to the following one, when you introduce the problem of $N_2O$ emissions.

**A:** Done!

10. L.s 51-62 I think it's better to unite these two paragraphs in one. Otherwise, it results confusing since the topic discussed continues from the first to the second.

**A:** Done!

11. L. 63 **contributes** Probably some words are missing here.

**A:** Done!

**12.** L. 63 f**or biological production as a N fertiliser** This seems confusing to me. Could you rephrase it?

**A:** We have clarified the sentence. The revised sentence is "Dissimilatory nitrate reduction to ammonium (DNRA) supplies $NH_4^+$ to the soil, conserves bioavailable N and prevents the leaching of $NO_3^-$ (Bai *et al*., 2020; Pandey *et al*., 2020).

**13.** L. 65 **both requiring $NO_3^-$** I don't think it is necessary to repeat this here again.

**A:** Done!

**14.** L. 68 **clad** There is a spelling mistake here.

**A:** Done!

**15.** L. 71 **the Hatch-Slack pathway** This phrase requires to be included in commas.

**A:** Done!

**16.** L. 71 **maise** This is a spelling mistake.

**A:** Done! We have corrected this error throughout the entire manuscript.

**17.** L.s 83-84 Could you rephrase this part? It sounds confusing to me.

**A:** We have revised the sentence followingly: "The general objectives of the study were to evaluate temporal patterns of gaseous N loss, link N-cycle processes with abundances of functional N cycle genes in arable mineral soil, and evaluate the performance of different crops (including novel crop in Northern Europe) in terms of biomass production and $N_2O$ emissions under mineral and organic fertilisation."

**18.** L. 86 **in arable mineral soil** What do you mean by "mineral soil"?

**A:** Mineral soil is typically defined to have less than 12% organic carbon in topsoil. Soil in our experiment is definitely classified as mineral soil. We have now removed the word "mineral" from the sentence to avoid confusion. Also, in methods we have mentioned the soil type, which indicates that it is a mineral soil.

**19.** L. 88 **decreases** This term here is not fitting.

**A:** Thank you for the comment! We have made changes, and the updated sentence fragment is "(3) in arable mineral soil, low soil moisture results in reduced $N_2O$ losses."

**20.** L. 88 **affects the soil microbial community** This hypothesis seems too general. Could you be more specific?

**A:** Yes, we agree with reviewer's comment. We have specified the hypothesis: "(4) amendment of manure fertiliser increases soil $N_2O$ emissions and affects the abundances of functional N cycle genes"

**21.** L. 89-90 This hypothesis does not sound very fitting to me for the purposes of the paper. I think it might be better to focus on the performance of this crop in comparison to the others. Otherwise, please provide further insights.

**A:** We agree with reviewer`s comment that hypothesis "sorghum is a prospective crop to cultivate in temperate climate" is not very fitting for the purposes of the paper and could not fully proven with presented results. We excluded this hypothesis from the paper.

**22.** L. 98 It might be better to move this sentence to the section preceding the description of the crop rotation.

**A:** Done!

**23.** L.s 104-109 In my opinion, this is the biggest problem of the whole manuscript. Although you specify there are three replicates per treatment, only one is present. This would be a major issue, if it wasn't a long-term field experiment. Therefore, please specify which crops have preceded in the few years before the described experiment started. Also, please specify if fertilization has been applied in the previous years and its entity.

**A:** Its long-term three-field crop rotation experiment with split-plot design to study effect of mineral and organic fertilisers established in 1989. Crop rotation order: spring wheat – spring barley – sorghum (potato before 2019). All fertilizer treatments have applied continuously from first harvest year in 1990. Manure is applied in every third year to potato/sorghum.

**24.** In addition, it would be ideal if you could provide further insights about the farmyard manure composition and whether it has been treated somehow (composting or something else).

**A:** The farmyard manure is cattle dung with straw bedding, freely fermented before use 6-8 months in heap. We have added this information to the Material and Methods section. Additionally, we have added chemical properties (C, N, P, K, dry matter) of the last manure amended in year 2022 and also last ten year average chemical properties of the manure in the Supplementary materials.

**25.** L. 104 I do not understand where the three replications are. A replicate is a treatment group to which the same levels of factors tested were applied in a way that allows to account for environmental factors' variability. Based on the experimental design you have provided, it seems that you only have one replicate per treatment group. If this is not the case, I think you only have pseudo-replicates.

**A:** We will include the use of a linear mixed-effects model in the revised version of the manuscript.

**26.** L.s 107-108 Manure was applied only to sorghum. I think you should make it clear also from the text.

**A:** Done! We have added following sentence to the revised version of the manuscript: "The farmyard manure treatment was applied only to sorghum."

**27.** L.s 113-114 As further specified in my comment on the text, it is better to clarify that additional N was applied with manure in the sorghum plots.

**A:** Done!

**28.** L. 125 **electron capture and flame ionisation detectors** Probably it is better to provide more details about these two instruments.

**A:** Done!

**29.** L.s 129-130 Please provide more details on the soil sampling process implemented (number of samples, rhizosphere or bulk soil).

A: Three auger samples from each point were collected for one composite sample for chemical and microbiological analyses. We have added the information to the Material and Methods section.

**30.** L.s 131-132 What are the instruments used for these analyses?

**A:** We have added the information to the Material and Methods section.

**31.** L. 134 Could you provide the reference for this method?

**A:** Done! We have provided reference to the Dumas method.

**32.** L.134 There is no need to use the full term here. It is better to use "C".

**A:** Done!

**33.** L.s 131-132 This paragraph should be united with the preceding one. Separating paragraphs discussing the same topic results in difficulty for the reader to understand their meaning.

**A:** Done! We have consolidated the paragraphs.

**34.** L. 146 I have some doubts about the term "total". In fact, you sampled roots up to a depth of 18 cm, but all these species' roots can easily reach lower depths.

**A:** With "total" we mention that above- and below-ground both were accounted. We agree that with the sampling method used some minor portion of root biomass might not be considered, but this would be a limited and probably negligible part.

**35.** L. 148 **maturity phase** Could you provide insights about the date when the measurement was done for each species?

**A:** The crops were harvested at maturity phase, and then state that biomass (both above- and belowground) was sampled on the harvest day.

**36.** L. 150 **Frasier et al. (2018)** This reference is not reported in the cited literature.

**A:** Thank you for pointing it out! We have added the missing reference.

**37.** L. 171 **extracted DNA** What is the DNA concentration used?

**A:** The DNA concentration was usually in the range of 20-35 ng/µl, although some samples had slightly lower or higher concentrations.

**38.** L.s 171, 172 **ml** I really think that here you mean microliter.

**A:** Done!

**39.** L.s 175-176  Could you specify which are the standard curve ranges used?

**A:** Yes, we can. We used the standard range of $10^4$ to $10^8$ for the bacterial 16S rRNA gene. We used the standard range of $10^4$ to $10^8$  for the archaeal 16S rRNA gene. We used the standard range of $10^2$ to $10^8$ 4for the archaeal amoA gene.  We used the standard range of 25 o $10^2$ for the bacterial amoA gene.  We used the standard range of 50 o $10^3$ for the comammox amoA gene. We used the standard range of $10^3$ o $10^5$ for the nirK gene.  We used the standard range of $10^4$ o $10^6$ for the nirS gene.  We used the standard range of 50 to$10^3$ for the nosZI gene and of $10^6$ to $10^8$ for nosZII gene.

**40.** L.180 **analysis** Since it is plural, it would be "analyses".

**A:** Done!

**41.** L. 183 **Analysis of variance (ANOVA)** Probably here it is necessary to specify the type of ANOVA used. Please provide me with your opinion.

**A:** Three-way ANOVA with the factors crop, fertilization rate and manure addition.

**42.** L. 185 **and** This "and" should be substituted by a comma.

**A:** Done!

**43.** L.s 185-187 Later, in the results, you use a different terminology to refer to this term.

**A:** Yes, we agree. We have unified the terminology in the manuscript to avoid misunderstanding.

**44.** L. 193 2017 Parentheses are missing here.

**A:** Done!

**45.** L.s 199-200 **in the soil** It is redundant to repeat this phrase.

**A:** Done!

**46.** L. 207 **Over all** Probably a different conjunction here would make the discussion more fluent.

**A:** Done!

**47.** L. 221 **increased** The form of this verb does not seem correct. Probably it is better to say that they "caused an increase in".

**A:** Done!

**48.** L. 226 **$p < 0.05$** This should be written without spaces.

**A:** Done!

49. L.s 226-227 For biomass yield, the hierarchy of uppercase letters seems to be B>C>A while for N content it seems to be C>D>B>A for yellow bars and A>B>C for the other two colours. I am right? If so, please correct the graph.

**A:** Yes, this is right. We have changed the hierarchy of uppercase letters to the same order.

50. L. 234 In table S3, $N_2$ emissions are just an estimation based on N2O emission if I'm not wrong. If this is the case, please specify it, otherwise it would be misleading.

**A:** Done! We have specified it.

51. L.s 240-246 Could you please specify in the text which of these differences are significant? It would also be ideal if you could specify the significance of the results with different letters in the table.

**A:** Done!

52. L. 264-265 Could you rephrase this? It seems quite redundant.

**A:** Done!

53. L. 286 **The effect of mineral N fertilisation** It would be better to write "mineral N fertilization effect".

**A:** Done!

54. L. 287 t**o effects of crop type** The correct form would be "to the effects of crop type".

**A:** Done!

55. L. 299 **Feature selection algorithm** Please align the name of this methodology with the one described in the methodology.

**A:** Done!

56. L. 301 **change** I'm not sure about the term "change". Probably "variations" or "alterations" are better.

**A:** Done! We have substituted the term "change" with "variations" throughout the entire manuscript.

57. L.s 316-325 I'm not sure if it is really necessary to describe all the significant correlations. Probably it's better to just choose the relevant ones.

**A:** Done!

58. L.s 333-334 Please, make it clear that you are specifying this in support of what previously stated.

**A:** Done!

**59.** L. 342 **indicates the dominance of nitrification over denitrification in N$_2$O-producing processes** Reference is lacking for the relevance of this ratio between the two genes.

**A:** We have provided reference for this ratio.

**60.** L.s 350-351 What does this mean?

**A:** We have further explained the Discussion section.

**61.** L. 359 **important** This term is not correct in this context.

**A:** Done!

**62.** L.s 362-363 A reference is lacking.

**A:** Done!

**63.** L.s 376-388 There is no need to separate this section in two different paragraphs, as the topic discussed is the same. Furthermore, this recurred also in other parts of the manuscript. It would be better to avoid writing short paragraphs with just a few sentences while separating the same topic in multiple paragraphs.

**A:** Done!

**64.** L. 386 **manure enhances the activity of soil microbes** It is better to not directly give this conclusion. It would be ideal if you propose this as one of the possible hypotheses.

**A:** We have changed the wording of the sentence and proposed it as one of the possibilities.

**65.** L.s 389-390 **with emissions increasing slower than linearly with the fertilisation rate** I don't think that this phrase describes correctly the observed trend.

**A:** Yes, we agree. We have removed it.

**66.** L.s 390-391 Are you sure this conclusion can be derived from just three points you have? L. 394 **N$_2$O emissions often grow exponentially when the applied N exceeds the necessary amount for crops** I don't understand what you mean by "often". I think that whether the growth is exponential or not depends on the number of N doses tested and their entity. I'm not sure about this so please let me know.

**A:** Yes, we agree with the reviewer's comment. We have removed sections from the manuscript that refer to a linear, exponential or any other response between fertilisation rate and N$_2$O emissions.

**67.** L. 397 **positive linear response** Here the same comment as before applies. Please provide further explanation about the linear or exponential response.

**A:** As mentioned in the previous comments, we have removed sections from the manuscript that refer to a linear, exponential or any other response between fertilisation rate and N$_2$O emissions.

**68.** L.s 400-405 I don't think that discussing this part can provide useful insights to this work.

**A:** Thank you for the comment!.We have removed the paragraph.

**69.** L. 406 **indirectly affecting** Probably it is better to say: "as it directly affects".

**A:** Done!

**70.** L. 410 Based on my personal experience, this information does not seem very fitting. Could you please check that it is correct?

**A: We checked and elaborated on the possible reasons further, and changed it accordingly.**

**71.** L.s 411-412 This sentence has no reference, and it is not very clear as it is not specified how water scarcity might enhance N gases emissions.

**A:** We have added suitable references and specified the link between gaseous N emissions and water scarcity.

**72.** L. 414 **$N_2O$ management should align with crop yield** It is not very clear here what is intended for "align".

**A:** Yes, we agree that it needs clarification. We mean that $N_2O$ management should be coordinated with futher crop yield objectives to sustain fastly growing human population. We have rephrased the sentence.

**73.** L. 414 Biomass It would be better to say "biomass production".

**A:** Done!

**74.** L. 416 **fertilisation rate 160 kg N ha−1** "of" is missing here.

**A:** Done!

**75.** L.s 416-417 **but our study shows increasing N2O emissions at higher fertilisation rates (Figure 6A), suggesting potential overfertilisation.** I don't understand what the connection with the previous sentence is.

**A: We have rephrased it.**

**76.** L.s 419-421 If this sentence is added to sustain the previously discussed results, I think it is better to specify it. Otherwise, it does not seem very clear.

**A:** Done!

**77.** L. 422 **rate 160 kg N ha−1** "of" is missing here.

**A:** Done!

**78.** L. 422 **increase** Probably it is better to use the past tense, as you are describing results you observed.

**A:** Done!

**79.** L. 423 **The fertilisation rate 80 kg N ha−1** "of" is missing here.

**A:** Done!

**80.** L.s 433-434 There are some typos in this sentence.

**A:** Done!

**81.** L.s 437-438 As you studied the abundance of microbial functional groups based on a DNA approach, I think that this speculation is too strong. What you can say is that the nitrification potential was higher than the denitrification one, but not that one process prevailed over one other.

**A:** We agree that it might be too strong to use word "dominance."We have corrected this throughout the entire manuscript. For example, we included the following sentence in the Conclusions section:

"$N_2O$ emissions were mostly caused by nitrification, with potential contribution from denitrification, comammox and DNRA processes."

We included following sentence in the Discussion section: "The significant positive correlation between the ratio of *amoA*/*nir* and $N_2O$ emissions ($\rho$= 0.20, p<0.001) indicates that nitrification potential was higher than denitrification potential and thereby $N_2O$ emissions were mainly related to nitrification in the soil."

**82.** L. 438 **N cycle** A hyphen is required.

**A:** Done!

**83.** L.s 442-443 **fertilisation rate 80 kg N ha−1** "of" is missing here.

**A:** Done!

**84.** L. 444 **positioning** This term is not very fitting here. Please, change it.

**A:** Done!

---

## Author Response (AR1)

Response to Anonymous Referee #1

**Reviewers' comments:**

**Anonymous Referee #1:** The manuscript discusses the effects of mineral N fertilization rates and manure amendment on soil $N_2O$ emissions across barley, wheat, and sorghum. The study leverages soil physicochemical analysis, nitrification- and denitrification-related genes, and gas emissions to assess the impact of fertilization strategies on N use efficiency and $N_2O$ emissions. The long-term aspects of the study site, the monthly variations of $N_2O$ emissions across crops, and the genetic components of N transforming pathways, provide a rich, publishable study. However, although the data is interesting, I do think this manuscript is not in good shape yet.

**A:** Thanks for the reviewer's thorough and constructive feedback. We have thoroughly reviewed comments and addressed all the critical points raised by the reviewer. Detailed responses are provided below.

1. The manuscript's writing style, particularly in the Introduction and Discussion sections, is disjointed and verbose. The introduction is wordy with some redundancies. For example, the first and second paragraphs could be consolidated.

**A:** Thank you for highlighting this. We have strengthened the Introduction and Discussion sections. We have shortened the text by modifying and removing less relevant sentences or sentence fragments. We have additionally included sentences into both sections (for example added factors that influence N-cycle processes). Also, we have consolidated some paragraphs in both Introduction and Discussion sections.

For example, we have consolidated first and second paragraph in Introduction section as reviewer suggested and removed less relevant sentences: "*The rising demand for agricultural commodities and the management of agroecosystems are important factors contributing to global environmental problems. Increasing crop yield while reducing pollution from agricultural production is crucial (Abdalla et al., 2019; Tilman et al., 2011). Global food demand projections suggest a 50% increase in agricultural production by 2050 (compared to 2012) to feed the fast-growing human population (FAO, 2017). Enhancing agricultural production involves actions such as expanding agricultural land, applying more fertilisers, and using water resources and fertilisers more effectively (Tian et al., 2021). In today's agricultural practises, the applied N with fertilisation is often excessive for plant needs (Robertson and Vitousek et al., 2009; Zhou et al., 2016). About half of the applied N to the fields is not taken up by crops (Coskun et al., 2017); which may lead to N loss in the surrounding environment. Main soil nitrogen loss mechanisms include denitrification, ammonia oxidation, N leaching, erosion of soil and ammonia ($NH_3$) volatilisation (Thomson et al., 2012). This results in adverse ecological impacts, such as eutrophication of aquatic ecosystems and increased gaseous emissions of N into the atmosphere (Cameron et al., 2013; Liu et al., 2017; Whetton et al., 2022).*"

In Discussion section, we have removed the repetitive parts and improved flowing of the text. We have also consolidated paragraphs, where discussion of the same topic was previously divided into multiple, short paragraphs.

2. The factors that influence nitrification, denitrification, comammox, and DNRA should be provided.

**A:** We have made adjustments, and the introduction now incorporates the factors that influence the above-mentioned processes. We have included in the Introduction section the following paragraphs:

"*Synthetic fertilizers containing $NH_3$ offer an immediate substrate for ammonia oxidizers, thus accelerating the nitrification process (Ayiti & Babalola, 2022). Also, fertilizers that raise soil pH can significantly enhance the nitrification rate, as increasing soil pH from 4.8 to 6.7 can boost nitrification rates by 30 times (DeForest & Otuya, 2020).*"

"*Carbon to nitrogen ratio (C/N) and $C/NO_3^-$ are recognised as the main environmental factors controlling, which nitrate-reducing process is favoured as DNRA and denitrifying microbes compete for $NO_3^-$ and carbon sources (Bai et al., 2020). DNRA is dominant in the presence of a high C/N ratio and low $NO_3^-$ availability, while the denitrification process favours a low ratio of C/N and $C/NO_3^-$ (Bai et al., 2020; Pandey et al., 2020).*"

3. Background information about effects of different crops on $N_2O$ emissions should be provided.

**A:** Thank you for drawing attention to this. To our understanding, there is a limited number of studies where comparisons between crop species and $N_2O$ emission have been made. We have included additional information, and now the Introduction section contains details about the effects of crops on $N_2O$ emissions. We have included the following paragraph in the Introduction section:

"*Only a limited number of studies have compared $N_2O$ emissions between different crop species. Abdalla et al. (2022) found that crop type has significant effect (p<0.05) on the BNE values from soil. Furthermore, Bouwman et al. (2002) also found that crop type has a significant influence on $N_2O$ emissions. However, study including 372 sites showed that cover crops did not have significant (p>0.05) effect on direct $N_2O$ emissions (Abdalla et al., 2019).*"

4. Additionally, the manuscript should clarify why nitrification is considered a primary source $N_2O$ fluxes. Though nitrification is the dominant step over denitrification in the soil, $N_2O$ is not the major product of nitrification.

**A:** In arable soil, nitrification tends to be a more important process than denitrification as arable soils are usually well-aerated and have sufficient oxygen to support nitrifying microbes. Yes, we agree that this part needs further clarification. In the manuscript, we mean that nitrification potential was higher than the denitrification one. We have rephrased the sentences considering this matter throughout the manuscript in the Abstract, Discussion and Conclusions section. For example, we included the following sentence in the Conclusions section:

*"N$_2$O emissions were mostly caused by nitrification with potential contribution from denitrification, comammox and DNRA processes."*

5. Hypothesis 5 (do you really need that many hypotheses?) is not a testable/measurable hypothesis, and how would the authors define "prospective"? Adaptability? Yield? N use efficiency? Water use efficiency?

**A:** We have reduced the number of hypotheses. We agree with reviewer`s comment that hypothesis "*sorghum (Sorghum bicolor x Sorghum sudanense) is a prospective crop to cultivate in temperate climate*" could not be fully proven with presented results. We excluded this hypothesis from the paper.

6. My major concern in M&M is the experimental designs. It's more like a pseudo-replicated (the three study plots within each crop are not independent) instead of a completed randomized block design by looking at Fig 1.

**A:** We used linear mixed-effects models in the revised version of the manuscript to overcome the possible problem of the pseudo-replicates. We used it to test statistical differences between N emissions of different fertilisation rates in plots with different crop types. We used spatial (different fertilisation rate) and temporal (sampling dates) effects as random effects. This model will help account for both fixed and random effects inside the experimental design, which provides better analysis of data.

7. Besides, the authors should consider providing more information about manure amendments, like the major source, the CN ratio of the manure, and whether the manure application is just one-time for this experiment or it's a part of long-term experiments (if so, the manure application started since which year?

**A:** Study is made on long-term three-field crop rotation experiment established in 1989 and all fertilization treatments are applied continuously from start. Manure treatment is amended with solid farmyard manure (ca 40 t ha$^{-1}$) in every third year before sorghum/potato. The farmyard manure is cattle dung with straw bedding, freely fermented before use 6-8 months in heap. We have added this information to the Material and Methods section. Additionally, we have added the average C, N, P and K content in manure added in year 2022 and during the last ten years in Table S1 in the Supplementary materials.

8. A climate diagram or bi-weekly/monthly precipitation amount should also be provided to align with soil moisture (Fig. S6) and N$_2$O emissions plots (Fig. 5).

**A:** Thank you for the comment! We have included a climate diagram of the study period (April-October 2022) in Figure S1 in the Supplementary materials section.

9. Other soil properties like pH, texture, and bulk density, which influence nutrient dynamics and gas emissions, are also crucial and should be included in the study.

**A:** The soil type is *Stagnic Luvisol* combined with *Fragic Glossic Retisol.* Texture by FAO classification is sandy loam: 57.86% sand (>0.063 mm), 33.58% silt (0.063–0.002 mm) and 8.55% clay (<0.002 mm). Soil bulk density was in range of 1.5 to 1.6 g cm$^{-3}$ with slightly lower values

for manure treatment plots. The average pH levels in spring 2022 were 5.4 for barley plots, 5.3 for wheat plots, 5.6 for sorghum plots without manure amendment, and 6.2 for sorghum plots with manure amendment. We have added the above-mentioned information to the Material and Methods section.

10. The author should report soil organic C instead of total C.

**A:** Soil organic carbon is already included in the analyses; it is just named a little differently because of the method used: HWEOC – hot-water extractable organic carbon. We can rename it to SOC if the reviewer requests.

11. I don't understand why the authors use PCA instead of simpler methods like bar charts to present soil C, N, and inorganic N in different sites.

**A:** PCA gives a good overview of the data and helps to observe trends and patterns. For example, lines 200-203: "*Fertilised plots had higher soil $N_{tot}$, $C_{tot}$, $NO_3^-$-N and $NH_4^+$-N content compared to non-fertilised plots according to the principal component analysis (PCA) (Figure 2). For sorghum without manure amendment plots (Figure 2C), $NO_3^-$-N and $NH_4^+$-N contents were more different from each other compared to sorghum with manure amendment plots, where $NO^-_3$-N and $NH_4^+$-N contents were relatively similar (Figure 2D).*"

However, as the reviewer requested, we have added additional figure (Figure S3) in the Supplementary materials section to represent TN and TC data in different sites.

12. I also think the authors should consider using other approaches (like structural equation models or approaches that can consider contributions from multiple factors) in addition to ANOVA and Pearson correlation to analyze their data. N cycling is complicated and has been influenced by many factors including vegetation, texture, soil moisture (precipitation), temperature, soil fertility and C concentration, and management practices like tillage, fertilization, etc. Simple correlation analysis may not always be the best way to capture those complicated interactions. And since the authors measured $N_2O$ emissions with time, I think they should analyze the data by different time periods instead of just cumulative fluxes.

**A:** Yes, we agree that SEM would have been a good option, and that is why we tested it as well. We tried SEM with a small set of soil chemical parameters (the ones that are the main substrates/controllers of nitrification and denitrification, adding nitrification and denitrification as latent variables), but the results were not clear. Additionally, we included more complicated modeling through linear mixed-effects modeling.

13. It's also odd for me to compare sorghum + manure with barley/wheat without manure application in Fig 6. It should be barley vs sorghum vs wheat as one part, and sorghum vs sorghum w/ manure as another part.

**A:** We have chosen to present all treatments and crop types together in Figure 6, because it provides an overview of the differences across all crop types and treatments. Additionally, we wanted to

limit the number of figures, as we have already seven figures in the main text and eight figures in the Supplementary materials section.

14. The Discussion sections probably need some major work. There are many repetitive parts ($N_2O$ emissions increased with high mineral N application) and many statements are contradicted with each other in the current version. For example, the authors said there is no correlation between soil moisture levels and $N_2O$ emissions or functional marker gene abundances. Then the authors note that the lowest levels of $N_2O$ emissions and functional marker gene abundances occurred during periods of low soil moisture.

A: In the Discussion section, we removed repetitive content and improved the flow of the text. Additionally, we consolidated paragraphs where the discussion of the same topic was previously divided into multiple short paragraphs.

We were not able to see any correlations between soil moisture levels and $N_2O$ emissions. Although we were able to see visually low $N_2O$ emissions in July in Figure 5 in the main text and low soil moisture content in July in Figure S4 in the Supplementary materials section. Also, we were able to see low gene abundances in some genes in Figures S5, S6, S7 and S8 in the Supplementary materials section.

15. Another example is the authors said no significant influence of crop type on $N_2O$ emissions, then the authors suggested sorghum as a potential crop in Northern Europe as sorghum maintained low $N_2O$ emissions.

A: Yes, there was no significant influence of crop type on $N_2O$ emissions. Probably the effect of fertiliser was much greater than the effect of crop type, therefore it was not visible. The suggestion is based on the smaller N losses compared to other two crop types in table S6 in the Supplementary materials section. We have removed the sentence, which suggested positioning sorghum as a potential crop for Northern Europe.

16. The authors also said $N_2O$ emissions increased with fertilization rates for wheat and barley plots, but the statistical results in Fig 6 showed no significant differences between N0 and N80.

A: Indeed, there is no significant differences between fertilisation rates N0 and N80 for wheat and barley plots in Figure 6A. However, there is a significant difference between fertilisation rates N0 and N160 in barley and wheat plots. Also, we can see significant differences between rates N80 and N160 for both barley and wheat plots.

17. I also don't know how to use the ratio of gene copy numbers to infer the resources of $N_2O$. Both nitrification and denitrification contribute to $N_2O$ emissions. "dominance" might overstate the results given the weak strength of the correlation. Nitrification and denitrification are complex processes influenced by a variety of environmental and microbial factors. This correlation alone does not conclusively establish dominance or any cause-effect relationship.

**A:** The *amoA*/*nir* ratio indicates the potential activity of nitrifying and denitrifying microorganisms as it shows abundance of these microbes. We agree that it might be too strong to use word "dominance." We have corrected this throughout the entire manuscript. For example, we included the following sentence in the Conclusions section:

"*$N_2O$ emissions were mostly caused by nitrification, with potential contribution from denitrification, comammox and DNRA processes.*"

18. It always needs extra caution on suggestions replacing current crops with sorghum. Assuming that sorghum with enhanced biological nitrification inhibition properties could reduce $N_2O$ given the same levels of N as other crops (corn, wheat, barley, etc) is applied, how much grain demand could be met by sorghum when considering large-scale implementation of the practice? Instead, including sorghum in the existing crop rotation and understanding its subsequent effects on N dynamics seems a more practical approach.

**A:** Yes, we agree that it must be checked further to determine if it would be suitable for large-scale implementation. However, we were more suggesting that it could be a great alternative in the near future from the perspectives of climate and yield. It looks like that sorghum`s subsequent effects on N dynamics seems like a good option. We have smoothed our wording.

19. Please use upper case L to represent liter.

A: We corrected it.

20. L52: 70% of N fertilizers were lost due to nitrification and denitrification? You just said about half of applied N to the field is not taken by plants (L44). In addition, how about N leaching and volatilization?

**A:** Sorry for the confusion. We have rephrased it more clearly to avoid any misunderstanding. We have added information about other possible N losses pathways such as N leaching and volatilization in the Introduction section. The text has been revised as follows:

"*The key microbial processes leading to soil N loss are nitrification and denitrification (Thomson et al., 2012). In agriculture, N fertilisers added to the soil can be lost due to these processes (Saud et al., 2022).*"

"*About half of the applied N to the fields is not taken up by crops (Coskun et al., 2017); which may lead to N loss in the surrounding environment. Main soil nitrogen loss mechanisms include denitrification, ammonia oxidation, N leaching, erosion of soil and ammonia ($NH_3$) volatilisation (Thomson et al., 2012).*"

21. L94: IOSDV: Should put full name first and abbreviation in the parentheses.

**A:** We corrected it.

22. L96: on crop type? Did you mean on crop responses of various crops?

**A:** Yes, exactly. The sentence has been revised to make it clearer as follows:

*"The experiment was set up as a three-field crop rotation experiment in 1989 to investigate the long-term effects of mineral and organic fertilisation on crop responses of various crops and soil properties."*

23. Table S1:. Please express the unit of herbicide application as L ha-1. Should be the same order as other figures: barley - Sorghum – wheat

**A:** Thank you for pointing this out. We have made the changes accordingly in the Supplementary materials section.

24. L120: Ø: diameter? is this 65 L the entire volume of PVC collars+lid? Looks like the volume is not consistent across treatments due to chamber extension, which create another sources of variable

**A:** It was not possible to measure different crops with the same 65 L chamber because the height of the crop exceeded chamber size.

Yes, 65 L is the entire volume of polyvinyl chloride chamber with collar. Chamber extensions were used for some treatments of sorghum on four sampling days. As chamber extensions increase the total volume of the chamber, it is essential to adjust the calculations accordingly. On all four occasions, the use of chamber extensions is considered in the calculations.

25. L188: What's biomass yield produced? Biomass production?

**A:** We agree that the term is confusing. We have corrected the manuscript according to the reviewer`s suggestion.

26. Figures: The figures should be labeled in order in the Result section. Fig S5 comes first in this draft so it should be S1. Similarly, Fig S6 -> S2. And Figure S1-4 should be S3-6. And please use the correct format for unit, like using mg kg-1 instead of mg/kg

**A:** Thanks for the comment. We have now labeled all the figures accordingly.

27. L221: The unit of Y axis in Fig 3B is not concentration.

**A:** Accepted. The word "concentration" has been replaced with "content."

28. Fig 3a: Using ton ha-1 in the y axis may better align with the context

**A:** Using ton ha$^{-1}$ can also be a good choice. However, we chose to leave the unit unchanged for now but can make the adjustment if needed.

29. Fig 5: precipitation data should be provided.

**A:** As mentioned above in comment number 8, we have included a climate diagram for the study period (April-October 2022) in Figure S1 in the Supplementary materials section.

30. L267: The statement that cumulative barley

**A:** Unfortunately, we do not understand the reviewer`s comment. It seems that part of the sentence is missing.

31. Fig 6: I don't understand the reason for estimating $N_2$ emissions in this study.

A: The reason for estimating $N_2$ emissions in this study was done to understand more widely the N cycling, for example denitrification. By including the functional genes responsible for denitrification in the study, additional estimation of $N_2$ emissions can provide insights into the denitrification process.

32. Table 1: I am not sure if the reader needs to know sum of squares, means square, w2. And it's odd that manure amendment is significant for $N_2O$ emissions in Table 1 but not in Fig 6.

A: We have removed sum of squares, means square, $\omega^2$ as reviewer suggested. In Figure 6, there is simple testing of statistical significance between different fertilisation rates and crop types. However, in the table 1, we have considered interactions between $N_2O$ emissions and different factors and measured the effect size of fertilisation rate, manure amendment, and crop type on $N_2O$ emissions.

33. L334: It seems long-term manure application showed no significant difference in $NO_3$, $NH_4$, $N_2O$, and $N_2$? That is odd. In Fig 6A, soils under 231 kg N ha$^{-1}$ (N0 at sorghum w/ manure) treatment produce lower cumulative $N_2O$ compared to those under 80 kg N ha$^{-1}$ (N80 at sorghum). That required some explanations.

A: There was no statistical differences in $N_2O$ emissions between fertilisation rates N0 and N80 on sorghum with manure plots, although we can see the differences in Figure 6. Also, there is no statistically significant differences between sorghum with and without manure amendment on the same fertilisation rate in Figure 6 (Uppercase letters indicate comparisons of the same fertilisation rate over crop types).

34. L339: This paragraph needs further expended. We generally expect organic fertilization would increase SOC, total N, yield, and N2O due to direct C & N (both labile and recalcitrant) inputs. Same as L379, what's the potential reason for different results in this study and previous studies?

A: We have added additional explanations.

35. L381: there are no significant difference between sorghum and sorghum + manure in N0 and N160 (Fig 6A).

A: Manure addition was significant according to the Table 1 in the main text, although it was not significant within all the fertiliser addition rates (Figure 6A).

36. L434: if there is a liner response, the authors should provide p-value and r2

A: Thanks for pointing that out. We have removed sections from the manuscript that refer to a linear response between fertilisation rate and $N_2O$ emissions due to the comments of anonymous referee nr 3.

**Reviewers' comments:**

**Anonymous Referee #2:** The manuscript fits well with the SOIL aims and scope. In general, it is well-organized and presented. And I would recommend this paper for publication. I would just suggest some points that I hope may be useful for the authors.

**A:** Thanks for the nice words and guidance! We have deeply considered all the reviewer suggestions. Detailed responses are provided below.

1. From the general perspective, I have noticed that the authors have calculated the $N_2$ in addition to $N_2O$ in the results. However, they have not mentioned that data on the discussion. I was wondering if the authors could relate the data mentioned to the genes and the completion of the denitrification process, for example.

**A:** Thank you for bringing this to our attention. We have now discussed it in the Discussion section also. We have less data from modeling, which made it difficult to analyze them with the abundances of genes. However, we checked it again and have used statistical techniques to compare them as well.

2. L 89: Hypothesis (5) is not clear. Does it mean optimal?

**A:** Thank you for the comment. The hypothesis is stated as "*sorghum (Sorghum bicolor x Sorghum sudanense) is a prospective crop to cultivate in temperate climate.*" This hypothesis considers sorghum as a crop that maximizes agricultural productivity while minimizing resource use and environmental impact (e.g., $N_2O$ emissions, N losses). Thus, yes, it does mean optimal.

However, the hypothesis "*sorghum is a prospective crop to cultivate in temperate climate*" is a bit too general and thereby we have excluded this hypothesis from the manuscript as suggested by other reviewers.

3. L 120: I understand that the corresponding corrections where made for the calculations when using the chamber extensions.

**A:** Yes, the use of chamber extensions is considered in the calculations. The total volume of the chamber was updated in the calculations when the extensions were used. As chamber extensions increase the total volume of the chamber, it is essential to adjust the calculations accordingly.

4. L 130: It would be important to state how much time lasted from the soil sample collection until analysis.

**A:** Thank you for pointing this out. The samples for microbial analyses were analysed three months after the last sampling in a running order. The microbial samples were stored in the freezer at $-20$ ˚C, as soon as possible to stabilise the samples and prevent further microbial processes from happening before further analyses. Soil chemical analyses were done 5 months after the sampling.

Until chemical analyses, samples were stored at +4 ˚C. Critically labile elements were measured as soon as possible after the sampling.

5.  L 183 – L 185: Here it would be important to explain what data is parametric and what non-parametric as ANOVA is a parametric test and the Spearman's rank a non-parametric one.

**A:** Thank you for highlighting this. This is correct. We have now modified our statistical analyses according to the reviewer's suggestion to prevent any false positive or negative results.

Table 1 in the manuscript is based on ANOVA test as cumulative $N_2O$ emission values are meeting the assumptions of parametric test. Later, Kruskal–Wallis test, post-hoc Tukey HSD test and Spearman correlation methods were used as this data was not normally distributed and thereby did not meet the assumptions of parametric test. The gene abundance data and environmental factor data was also not normally distributed. We have added the above-mentioned information into the Material and methods section.

**Reviewers' comments:**

**Anonymous Referee #3:** This work presents a relevant topic, that is soil $N_2O$ emissions management, to understand N fertilization and crop type impact, along with a possible involvement of soil microbiome. Therefore, in my opinion, it is relevant to SOIL aims and scope. Introduction is well constructed, and it highlights the relevance of the study in a broader context. Some modifications in text structure are required. Hypotheses and objectives are clearly stated and coherent with the methodology used, even though some of them might require improvements.

**A:** Thank you for your valuable feedback. We have carefully reviewed your comments and addressed all the critical points raised. Detailed responses are provided below.

1. The biggest problem of this paper is a major lack in appropriately describing the experimental design. In fact, it is necessary to specify which crops immediately preceded the ones tested in the current experiment and which fertilization treatments they received. The reason why it is so important is that this experimental design would not be valid if this wasn't a long-term experiment, as there aren't multiple separate and randomized plots. In fact, I think that what you refer to as "replicates" are only multiple sampling points of one unique plot per treatment. If this lack in methodological information will not be addressed appropriately, I fear that it might seriously undermine the reproducibility of this work. In addition, some information about one of the treatments is lacking (manure amendment).

**A:** Thank you for pointing this out. We have revised the manuscript and added additional information about the experimental design according to reviewers' suggestions.

Study is made on long-term three-field crop rotation experiment established in 1989 and all fertilization treatments are applied continuously from start. Manure treatment is amended with solid farmyard manure (ca 40 t ha$^{-1}$) in every third year before sorghum/potato. The last year of manure amendment was in year 2022. The farmyard manure is cattle dung with straw bedding, freely fermented before use 6-8 months in heap. We have added this information to the Material and Methods section. Additionally, we have added the average C, N, P and K content in manure added in year 2022 and during the last ten years in Table S1 in the Supplementary materials. In the Material and Methods section, we already had the following information about the preceded crops: "Initially, the crop rotation was potato–spring wheat–spring barley (Astover *et al.*, 2016). In 2019, potato was replaced with sorghum-sudangras hybrid."

We have included the use of linear mixed-effects models in the revised version of the manuscript to overcome the possible problem of the pseudo-replicates. We used it to test statistical differences between N emissions of different fertilisation rates in plots with different crop types. We used spatial (different fertilisation rate) and temporal (sampling dates) effects as random effects. This model will help account for both fixed and random effects inside the experimental design, which provides better analysis of data.

2. Results have been described quite clearly, although sometimes too much detail is given about findings that don't have a wide importance. In some graphs I think there are some mistakes in results presentation. In the discussion section there are some problems related to the flowing of the text. In fact, often the description of the same topic is divided into multiple, short paragraphs, thus creating some confusion for the reader. Moreover, some of the speculations are too strong based on the presented results. Overall, the manuscript has a great potential to be improved, but only if the issue with the experimental design is correctly and extensively addressed, as it is the most serious problem of this work.

**A:** Thank you for the constructive feedback. We have revised the manuscript according to reviewer's feedback. In Discussion section, we have removed the repetitive parts and improved flowing of the text. We have also consolidated paragraphs, where discussion of the same topic was previously divided into multiple paragraphs. We have also revised the figures and the Results section. In addition, we smoothed the text where needed.

3. L. 12: $N_2O$ I think the term emission is missing here.

**A:** Done!

4. L.s 16-17 You mean higher compared to the application of mineral fertilizers?

**A:** Yes, we mean in comparison to mineral fertilisation. We have clarified it in the manuscript.

5. L. 19 **Microbial analyses** Could you be more specific here?

**A:** Thank you for pointing this out. We have revised the sentence followingly: "*Quantification of nitrogen cycle functional genes also showed the potential role of denitrification, comammox and DNRA processes as a source of $N_2O$.*"

6. L. 23 **sorghum** It is not necessary to repeat the term sorghum here again.

**A:** Done!

7. L. 33 **maise** There is a spelling mistake here.

**A:** Done! We have corrected this error throughout the entire manuscript.

8. L. 34 **in present agricultural regions due to climate system changes** I am sorry, but this is not very clear to me. Could you please rephrase it.

**A:** We have removed this sentence from the Introduction section to avoid too verbose text and redundancies.

9. L. 38 This sentence does not fit well in this paragraph. I think it is better to move it to the following one, when you introduce the problem of $N_2O$ emissions.

**A:** Done!

10. L.s 51-62 I think it's better to unite these two paragraphs in one. Otherwise, it results confusing since the topic discussed continues from the first to the second.

**A:** Done!

**11.** L. 63 **contributes** Probably some words are missing here.

**A:** Done!

**12.** L. 63 f**or biological production as a N fertiliser** This seems confusing to me. Could you rephrase it?

**A:** We have clarified the sentence. The revised sentence is "Dissimilatory nitrate reduction to ammonium (DNRA) supplies $NH_4^+$ to the soil, conserves bioavailable N and prevents the leaching of $NO_3^-$ (Bai *et al.*, 2020; Pandey *et al.*, 2020).

**13.** L. 65 **both requiring $NO_3^-$** I don't think it is necessary to repeat this here again.

**A:** Done!

**14.** L. 68 **clad** There is a spelling mistake here.

**A:** Done!

**15.** L. 71 **the Hatch-Slack pathway** This phrase requires to be included in commas.

**A:** Done!

**16.** L. 71 **maise** This is a spelling mistake.

**A:** Done! We have corrected this error.

**17.** L.s 83-84 Could you rephrase this part? It sounds confusing to me.

**A:** We have revised the sentence followingly: "*The general objectives of the study were to evaluate temporal patterns of gaseous N loss, link N-cycle processes with abundances of functional N cycle genes in arable soil, and evaluate the performance of different crops (including novel crop in Northern Europe) in terms of biomass production and $N_2O$ emissions under mineral and organic fertilisation.*"

**18.** L. 86 **in arable mineral soil** What do you mean by "mineral soil"?

**A:** Mineral soil is typically defined to have less than 12% organic carbon in topsoil. Soil in our experiment is definitely classified as mineral soil. We have now removed the word "mineral" from the sentence to avoid confusion. Also, in methods we have mentioned the soil type, which indicates that it is a mineral soil.

**19.** L. 88 **decreases** This term here is not fitting.

**A:** Thank you for the comment! We have made changes, and the updated sentence fragment is "*(3) in arable soil, low soil moisture results in reduced $N_2O$ losses.*"

**20.** L. 88 **affects the soil microbial community** This hypothesis seems too general. Could you be more specific?

**A:** Yes, we agree with reviewer's comment. We have specified the hypothesis: "*(4) amendment of manure fertiliser increases soil $N_2O$ emissions and affects the abundances of functional N cycle genes.*"

21. L. 89-90 This hypothesis does not sound very fitting to me for the purposes of the paper. I think it might be better to focus on the performance of this crop in comparison to the others. Otherwise, please provide further insights.

**A:** We agree with reviewer`s comment that hypothesis "*sorghum is a prospective crop to cultivate in temperate climate*" is not very fitting for the purposes of the paper and could not fully proven with presented results. We excluded this hypothesis from the paper.

22. L. 98 It might be better to move this sentence to the section preceding the description of the crop rotation.

**A:** Done!

23. L.s 104-109 In my opinion, this is the biggest problem of the whole manuscript. Although you specify there are three replicates per treatment, only one is present. This would be a major issue, if it wasn't a long-term field experiment. Therefore, please specify which crops have preceded in the few years before the described experiment started. Also, please specify if fertilization has been applied in the previous years and its entity.

**A:** Its long-term three-field crop rotation experiment with split-plot design to study effect of mineral and organic fertilisers established in 1989. All fertilizer treatments have been applied continuously continuously from start. Manure is applied every third year to potato/sorghum. Crop rotation order: spring wheat – spring barley – sorghum (potato was in sorghum plots before year 2019).

24. In addition, it would be ideal if you could provide further insights about the farmyard manure composition and whether it has been treated somehow (composting or something else).

**A:** The farmyard manure is cattle dung with straw bedding, freely fermented before use 6-8 months in heap. We have added this information to the Material and Methods section. Additionally, we have added the average C, N, P and K content in manure added in year 2022 and during the last ten years in Table S1 in the Supplementary materials.

25. L. 104 I do not understand where the three replications are. A replicate is a treatment group to which the same levels of factors tested were applied in a way that allows to account for environmental factors' variability. Based on the experimental design you have provided, it seems that you only have one replicate per treatment group. If this is not the case, I think you only have pseudo-replicates.

**A:** We have included the use of linear mixed-effects models in the revised version of the manuscript to overcome the possible problem of the pseudo-replicates. We used it to test statistical differences between N emissions of different fertilisation rates in plots with different crop types.

We used spatial (different fertilisation rate) and temporal (sampling dates) effects as random effects. This model will help account for both fixed and random effects inside the experimental design, which provides better analysis of data.

**26.** L.s 107-108 Manure was applied only to sorghum. I think you should make it clear also from the text.

**A:** Done! We have added following sentence to the revised version of the manuscript: "*The farmyard manure treatment was applied only to sorghum.*"

**27.** L.s 113-114 As further specified in my comment on the text, it is better to clarify that additional N was applied with manure in the sorghum plots.

**A:** Done!

**28.** L. 125 electron capture and flame ionisation detectors Probably it is better to provide more details about these two instruments.

**A:** Done!

**29.** L.s 129-130 Please provide more details on the soil sampling process implemented (number of samples, rhizosphere or bulk soil).

A: Three auger samples from each point (both bulk and rhizosphere soil were sampled) were collected for one composite sample for chemical and microbiological analyses. All in all, 216 samples were collected for chemical analyses and 144 samples for microbial analyses. We have added this information to the Material and Methods section.

**30.** L.s 131-132 What are the instruments used for these analyses?

**A:** We have added the information about instruments measuring soil temperature and moisture to the Material and Methods section.

**31.** L. 134 Could you provide the reference for this method?

**A:** Done! We have provided reference to the Dumas method in the manuscript. The method is based on the ISO standard No. 13878:1998 (Soil quality — Determination of total nitrogen content by dry combustion, also known as "elemental analysis").

**32.** L.134 There is no need to use the full term here. It is better to use "C".

**A:** Done!

**33.** L.s 131-132 This paragraph should be united with the preceding one. Separating paragraphs discussing the same topic results in difficulty for the reader to understand their meaning.

**A:** Done! We have consolidated the paragraphs.

**34.** L. 146 I have some doubts about the term "total". In fact, you sampled roots up to a depth of 18 cm, but all these species' roots can easily reach lower depths.

**A:** With "total" we mention that above- and below-ground both were accounted. We agree that with the sampling method used, some minor portion of root biomass might not be considered, but this would be a limited and probably negligible part.

**35.** L. 148 **maturity phase** Could you provide insights about the date when the measurement was done for each species?

**A:** The total biomass (both above- and belowground) was harvested and measured on the harvest day of each crop. We *have* added this information in the Material and methods section. The harvest days in the field are displayed in Table S2 in Supplementary materials.

**36.** L. 150 **Frasier et al. (2018)** This reference is not reported in the cited literature.

**A:** Thank you for pointing it out! We have added the missing reference.

**37.** L. 171 **extracted DNA** What is the DNA concentration used?

**A:** The DNA concentration was usually in the range of 20-35 ng/µl, although some samples had slightly lower or higher concentrations.

**38.** L.s 171, 172 **ml** I really think that here you mean microliter.

**A:** Yes, correct. Done!

**39.** L.s 175-176 Could you specify which are the standard curve ranges used?

**A:** Yes, we can. We used the standard range of $10^4$ to $10^8$ for the bacterial 16S rRNA gene, $10^6$ to $10^8$ for the archaeal 16S rRNA gene, $10^4$ to $10^6$ for the archaeal *amoA* gene, $10^3$ to $10^5$ for the bacterial *amoA* gene, $10^2$ to $10^4$ for the comammox *amoA* gene, $10^5$ to $10^7$ for the *nirK* gene, $10^4$ to $10^6$ for the *nirS* gene, $10^3$ to $10^5$ for the *nosZI* gene, $10^5$ to $10^8$ for *nosZII* gene and standard range of 50 to $10^3$ for the *nrfA* gene.

**40.** L.180 **analysis** Since it is plural, it would be "analyses".

**A:** Done!

**41.** L. 183 **Analysis of variance (ANOVA)** Probably here it is necessary to specify the type of ANOVA used. Please provide me with your opinion.

**A:** Three-way ANOVA with the factors crop, fertilization rate and manure addition.

**42.** L. 185 **and** This "and" should be substituted by a comma.

**A:** Done!

**43.** L.s 185-187 Later, in the results, you use a different terminology to refer to this term.

**A:** Yes, we agree. We have unified the terminology in the manuscript to avoid misunderstanding.

**44.** L. 193 2017 Parentheses are missing here.

**A:** Done!

**45.** L.s 199-200 **in the soil** It is redundant to repeat this phrase.

**A:** Done! We have removed the repetition.

**46.** L. 207 **Over all** Probably a different conjunction here would make the discussion more fluent.

**A:** Done! The have revised the sentence as follows: "*Soil moisture was not significantly linked to gene copy numbers across all crop types, except nirS.*"

**47.** L. 221 **increased** The form of this verb does not seem correct. Probably it is better to say that they "caused an increase in".

**A:** We have corrected the sentence as the reviewer suggested.

**48.** L. 226 **$p < 0.05$** This should be written without spaces.

**A:** Done!

**49.** L.s 226-227 For biomass yield, the hierarchy of uppercase letters seems to be B>C>A while for N content it seems to be C>D>B>A for yellow bars and A>B>C for the other two colours. I am right? If so, please correct the graph.

**A:** We have removed the uppercase letters from the figure due to limited number of observations in the case of total dry weight biomass and N content in total dry weight biomass. Linear mixed-effects model (LMM) was not possible to apply for statistical differences between different fertilisation rates and crop types.

**50.** L. 234 In table S3, $N_2$ emissions are just an estimation based on N2O emission if I'm not wrong. If this is the case, please specify it, otherwise it would be misleading.

**A:** Yes, we have specified it.

**51.** L.s 240-246 Could you please specify in the text which of these differences are significant? It would also be ideal if you could specify the significance of the results with different letters in the table.

**A:** We have specified the differences in the text.

**52.** L. 264-265 Could you rephrase this? It seems quite redundant.

**A:** We have rephrased the sentence.

**53.** L. 286 **The effect of mineral N fertilisation** It would be better to write "mineral N fertilization effect".

**A:** We corrected it as the reviewer suggested.

**54.** L. 287 t**o effects of crop type** The correct form would be "to the effects of crop type".

**A:** Done!

**55.** L. 299 **Feature selection algorithm** Please align the name of this methodology with the one described in the methodology.

**A:** We have unified the terminology in the manuscript to avoid misunderstanding.

**56.** L. 301 **change** I'm not sure about the term "change". Probably "variations" or "alterations" are better.

**A:** Done! We have substituted the term "change" with "variations" throughout the manuscript.

**57.** L.s 316-325 I'm not sure if it is really necessary to describe all the significant correlations. Probably it's better to just choose the relevant ones.

**A:** Yes, we agree. We have removed the less relevant significant correlations in the text.

**58.** L.s 333-334 Please, make it clear that you are specifying this in support of what previously stated.

**A:** Yes, we have specified it.

**59.** L. 342 **indicates the dominance of nitrification over denitrification in $N_2O$-producing processes** Reference is lacking for the relevance of this ratio between the two genes.

**A:** We have provided references for this ratio.

**60.** L.s 350-351 What does this mean?

**A:** We have further explained it in the Discussion section.

**61.** L. 359 **important** This term is not correct in this context.

**A:** Done!

**62.** L.s 362-363 A reference is lacking.

**A:** We have included references.

**63.** L.s 376-388 There is no need to separate this section in two different paragraphs, as the topic discussed is the same. Furthermore, this recurred also in other parts of the manuscript. It would be better to avoid writing short paragraphs with just a few sentences while separating the same topic in multiple paragraphs.

**A:** We have consolidated the paragraphs and have taken it into account in other parts of the manuscript as well.

**64.** L. 386 **manure enhances the activity of soil microbes** It is better to not directly give this conclusion. It would be ideal if you propose this as one of the possible hypotheses.

**A:** We have changed the wording of the sentence and proposed it as one of the possibilities.

**65.** L.s 389-390 **with emissions increasing slower than linearly with the fertilisation rate** I don't think that this phrase describes correctly the observed trend.

**A:** Yes, we agree. We have removed this sentence.

**66.** L.s 390-391 Are you sure this conclusion can be derived from just three points you have? L. 394 **N₂O emissions often grow exponentially when the applied N exceeds the necessary amount for crops** I don't understand what you mean by "often". I think that whether the growth is exponential or not depends on the number of N doses tested and their entity. I'm not sure about this so please let me know.

**A:** Yes, we agree with the reviewer's comment. We have removed sections from the manuscript that refer to a linear, exponential or any other response between fertilisation rate and N₂O emissions.

**67.** L. 397 **positive linear response** Here the same comment as before applies. Please provide further explanation about the linear or exponential response.

**A:** As mentioned in the previous comments, we have removed sections from the manuscript that refer to a linear, exponential or any other response between fertilisation rate and N₂O emissions.

**68.** L.s 400-405 I don't think that discussing this part can provide useful insights to this work.

**A:** Thank you for the comment! We have removed the paragraph as reviewer suggested.

**69.** L. 406 **indirectly affecting** Probably it is better to say: "as it directly affects".

**A:** Done!

**70.** L. 410 Based on my personal experience, this information does not seem very fitting. Could you please check that it is correct?

A: Thank you for pointing this out! We have removed the sentence.

**71.** L.s 411-412 This sentence has no reference, and it is not very clear as it is not specified how water scarcity might enhance N gases emissions.

**A:** We have revised it and added suitable references.

72. L. 414 **N₂O management should align with crop yield** It is not very clear here what is intended for "align".

**A:** Yes, we agree that it needs clarification. We mean that N₂O management should be aligned with future crop yield objective to increase yield and thereby support the rapidly growing human population. We have rephrased the sentence as follows: "*Considering climate changes and population growth, N₂O management should be aligned with the future need to increase crop yield and sustain rapidly increasing human population.*"

**73.** L. 414 Biomass It would be better to say "biomass production".

**A:** Done!

**74.** L. 416 **fertilisation rate 160 kg N ha⁻1** "of" is missing here.

**A:** Done!

**75.** L.s 416-417 **but our study shows increasing N2O emissions at higher fertilisation rates (Figure 6A), suggesting potential overfertilisation.** I don't understand what the connection with the previous sentence is.

**A:** We have rephrased it to avoid misunderstanding.

**76.** L.s 419-421 If this sentence is added to sustain the previously discussed results, I think it is better to specify it. Otherwise, it does not seem very clear.

**A:** Yes, we agree. We have specified it.

**77.** L. 422 **rate 160 kg N ha−1** "of" is missing here.

**A:** Done!

**78.** L. 422 **increase** Probably it is better to use the past tense, as you are describing results you observed.

**A:** Done!

**79.** L. 423 **The fertilisation rate 80 kg N ha−1** "of" is missing here.

**A:** Done!

**80.** L.s 433-434 There are some typos in this sentence.

**A:** We have corrected the sentence as follows: "*Additionally, the number of N-cycle genes that are significant in the variations of $N_2O$ emissions also increased with manure amendment.*"

**81.** L.s 437-438 As you studied the abundance of microbial functional groups based on a DNA approach, I think that this speculation is too strong. What you can say is that the nitrification potential was higher than the denitrification one, but not that one process prevailed over one other.

**A:** We agree that it might be too strong to use word "dominance." We have corrected this throughout the entire manuscript. For example, we included the following sentence in the Conclusions section:

"*$N_2O$ emissions were mostly caused by nitrification, with potential contribution from denitrification, comammox and DNRA processes.*"

**82.** L. 438 **N cycle** A hyphen is required.

**A:** Done!

**83.** L.s 442-443 **fertilisation rate 80 kg N ha−1** "of" is missing here.

**A:** Done!

84. L. 444 **positioning** This term is not very fitting here. Please, change it.

**A:** We have corrected the sentence and removed this term.

---

## Author Response (AR2)

Response to Topic Editor

**Topic editors' comments:**

**Topic editor:** As indicated by the reviwer, the authors have sucesfully addressed the comments and suggestions raised previouly. There is still a pending issue related to the linear mixed modelling introduced in this last revised version and a couple of minor edits that needs to be addresed before accepting the manuscipt.

**A:** Thanks for the feedback. We have taken into consideration all the critical points raised by the topic editor. Detailed responses are provided below.

**1.** Regarding the Linear mixed modelling, an in agreement with the reviwer, there is something that is not clear: The use of random variables in the modelling can help addressing the limitations of the experimental design to a certain degree: There is no statistics that will remove the pseudoreplication that is derived from the experimental design, since you have only 1 plot per type (crop x fertilization rate) and 3 sampling spots within the plot. In fact, the sampling spots might be less than 5 meters appart? So that is effectively a pseudoreplication.

However, experimental sites like this have their value, and we have to take them as they are. Regardless of this issue, using random variables can improve the robustness of the analyses. You indicate that sampling date was used as a random variable to account for repeated measurements, which is ok. But also that different fertilization rate was used to account for the spatial structure. The problem is that you cannot use the same variable as fixed and random effect. If you are testing for the effect of fertilization rate, how can you include it a random variable too? I wonder if what you used as random variable was actually the plot (n=12), so that the 3 sampling spots are grouped. This way the model would account for the spatial structure (12 plots, 3 spots per plots). You would need to clarify and correct this issue if needed.

It would also be adequate that the lack of proper replication, and thus the limitation of the study, is briefly acknowledged in the discussion.

**A:** We have revised the manuscript and have now used only temporal (sampling dates) effect as random effect in the linear mixed-effects model. We used it to test statistical differences between $N_2O$ emissions of different fertilisation rates in plots with different crop types. The experiment was organised into 12 plots in a systematic block design (Figure 1 in the manuscript) with three sampling spots per plot. We must take into consideration that replicates of sampling might not be statistically independent since the experiment had 1 plot per type (crop type and fertilisation rate) and three samples were taken within the same plot.

We have revised the Discussion by adding the limitations of the study to the text as follows:

"*However, it is important to consider that the sample replicates may not be fully statistically independent, as the experiment involved only one plot per treatment combination (crop type × fertilisation rate), with three samples taken within the same plot.*

**Reviewers' comments:**

**Anonymous Referee #3:** In this version of the manuscript "Interactions of fertilisation and crop productivity on soil nitrogen cycle microbiome and gas emissions", the authors have addressed most of the critical comments that were raised in the review of the previous version, thus improving its quality.

**A:** Thank you for the valuable feedback.

1. Anyway, there is one aspect, not mentioned in the preceding version, on which I would kindly ask for clarification. In fact, in contrast with the stated objectives of this work, and with the way results are presented, in this version of the manuscript it is specified that, for statistical analyses, "For $N_2O$ emissions and gene parameters, spatial (different fertilisation rate) and temporal (sampling dates) effects were used as random effects. For $N_2$ emissions, spatial effect (different fertilisation rate) was used as a random effect" (L.s 214-216). I don't understand why these factors would be considered as random effects and, for the way results were presented, I'm given to understand that probably they weren't. Therefore, I think there might be a possibility that this could be a mistake in the text. If so, I think it should be addressed. If this is not just a mistake in the text, in my opinion, there would be a contrast between statistical analyses and results. Therefore, I would kindly ask the authors for further explanation on this point.

A: We have excluded spatial (different fertilisation rate) effect as random effect from the linear mixed-effects model. Now, only the temporal effect (sampling dates) is used as a random effect in the linear mixed-effects model.

2. L. 292 "expect" I think that this is one is a typo.

**A:** Done! We have corrected this error.

3. L. 311 "genetic parameters" I don't think that this phrase is fitting in this context to describe the abundance of microbial groups.

**A:** Thank you for the comment! We have changed the subtitle as follows:

"*3.4 Relationships between environmental parameters, gene abundances and ratios, and N emissions*"

---

## Author Response (AR3)

Response to Topic Editor

**Topic editors' comments:**

**Topic editor:** The authors have correcly addressed most of the comments with few final details to address prior publication.

**A:** Thank you for the valuable feedback. Detailed responses are provided below.

1. Thanks for the revision. One (hopefully) last statistical doubt I would need to clarify: Manure was only applied to one of the crop types (Sorghum) however it is included in the ANOVA analysis as a treatment equivalent to to Fertilization rate. When doing so, you will be comparing the 3 plots of shorgum+manure vs the 9 plots of shorgum/Barley/Wheat. In my opinion that is not a correct approach to evaluate the effect of manure, as the crop type effect is also confounded there. I would suggest the manure effect is tested in a separate ANOVA for only the Sorghum crop type, so the comparison is 3 with manure vs. 3 without manure.

**A:** We have revised the manuscript and have now used only sorghum crop type to assess the effect of manure. A separate ANOVA has been conducted for the sorghum crop type, comparing three sorghum plots treated with manure to three sorghum plots without manure amendment. The results are presented in Table 1B in the main text.

In Figure 6A, we have now also included a comparison between three sorghum plots treated with manure and three sorghum plots without manure.

2. In addition, it would be better to have both the ANOVAs and LME results in tables inlcuded in the main text.

**A:** We have provided the ANOVA tables and LME results are also included in the main text.